# A Multi-Objective Optimization Problem Solving Method Based on Improved Golden Jackal Optimization Algorithm and Its Application

**DOI:** 10.3390/biomimetics9050270

**Published:** 2024-04-28

**Authors:** Shijie Jiang, Yinggao Yue, Changzu Chen, Yaodan Chen, Li Cao

**Affiliations:** 1School of Intelligent Manufacturing and Electronic Engineering, Wenzhou University of Technology, Wenzhou 325035, Chinayueyinggao2006@163.com (Y.Y.);; 2Intelligent Information Systems Institute, Wenzhou University, Wenzhou 325035, China

**Keywords:** golden jackal algorithm, tent map, sine–cosine algorithm, Cauchy mutation, UAV path planning

## Abstract

The traditional golden jackal optimization algorithm (GJO) has slow convergence speed, insufficient accuracy, and weakened optimization ability in the process of finding the optimal solution. At the same time, it is easy to fall into local extremes and other limitations. In this paper, a novel golden jackal optimization algorithm (SCMGJO) combining sine–cosine and Cauchy mutation is proposed. On one hand, tent mapping reverse learning is introduced in population initialization, and sine and cosine strategies are introduced in the update of prey positions, which enhances the global exploration ability of the algorithm. On the other hand, the introduction of Cauchy mutation for perturbation and update of the optimal solution effectively improves the algorithm’s ability to obtain the optimal solution. Through the optimization experiment of 23 benchmark test functions, the results show that the SCMGJO algorithm performs well in convergence speed and accuracy. In addition, the stretching/compression spring design problem, three-bar truss design problem, and unmanned aerial vehicle path planning problem are introduced for verification. The experimental results prove that the SCMGJO algorithm has superior performance compared with other intelligent optimization algorithms and verify its application ability in engineering applications.

## 1. Introduction

With the rapid development of artificial intelligence and industrial technology, the demand for algorithm performance is increasing day by day. Traditional optimization algorithms have found it difficult to meet the needs of the rapid progress of society [1]. In response to the challenges faced by mathematical optimization algorithms in dealing with large-scale and highly complex problems, a series of metaheuristic optimization algorithms have emerged [2,3]. These algorithms are deeply inspired by the laws of nature, species evolution, and the behavior of biological populations. In particular, swarm intelligence algorithms have attracted much attention from the academic community because of their simulation of the intelligent behavior among biological populations [4,5,6].

The application of swarm intelligence algorithms has widely permeated many fields such as medicine, e-commerce, unmanned aerial vehicle technology, and mobile robots [7,8]. The advancement of science and technology has driven the development of numerous swarm intelligence algorithms, including genetic algorithms (GAs), particle swarm optimization (PSO) [9], whale optimization algorithms (WOAs) [10], and sparrow search algorithm (SSAs) [11]. These algorithms have drawn the attention of numerous scholars due to their practicality, ease of use, and high efficiency, giving rise to many emerging optimization algorithms. For instance, Wang et al. proposed the monarch butterfly optimization (MBO) by observing the migration and adaptation behaviors of monarch butterflies [12]. Wang et al. derived the wolf pack search algorithm (WPA) from the hunting and survival behaviors of wolves [13]. Xue et al. proposed the sparrow search algorithm (SSA) by simulating the predation and anti-predation mechanisms of sparrows [14]. Faramarzi et al. were inspired by the behavior of predators in the ocean to develop the marine predators algorithm (MPA) [15], while Meng et al. proposed the chicken swarm optimization algorithm (CSO) by simulating the hierarchical system and foraging behavior within a chicken flock [16]. The GJO algorithm is a heuristic intelligence algorithm proposed by Chopra et al. in 2022, which emulates the cooperative foraging behavior of golden jackals and the use of multiple hunting strategies [17]. The GJO algorithm exhibits characteristics such as fewer parameters, a simple structure, and a certain search capability, thus receiving widespread attention [18,19]. However, the theoretical system of the GJO algorithm is not yet complete, and there are issues such as a slow convergence rate, low solution accuracy, a tendency to fall into local optima, and sensitivity to parameter settings [20].

Given the shortcomings of the GJO algorithm, it is of particular significance to enhance and improve it to boost its performance. To overcome the limitations of the GJO algorithm, this paper presents the sine–cosine and Cauchy mutation golden jackal optimization algorithm (SCMGJO), which integrates the sine–cosine algorithm and the Cauchy mutation. The main improvement strategies include introducing the reverse learning of the tent mapping to initialize the population, applying the sine–cosine strategy in the update of the prey position within the algorithm to enhance the diversity of the population in later iterations, avoiding premature convergence, improving the accuracy of optimization, and accelerating convergence; simultaneously, the Cauchy mutation strategy is utilized to assist the algorithm in escaping local optima and further accelerating convergence. Through comparisons with 23 classical benchmark functions and various performance test indicators alongside basic intelligent algorithms, as well as combined with the verification of engineering optimization design problems, the effectiveness and superiority of the SCMGJO algorithm are fully demonstrated.

## 2. Related Work

The main idea of swarm intelligence optimization algorithms is to simulate the foraging, reproduction, and other behaviors of some social species in nature, abstract these behaviors into various mathematical models, and select appropriate evaluation functions for evaluation. Since the proposal of the golden jackal optimization algorithm, it has attracted widespread attention from researchers, and many researchers in this field have devoted themselves to the study of the golden jackal optimization algorithm [21]. Some researchers are committed to researching the application of the golden jackal optimization algorithm in different fields, such as engineering optimization and intelligent control, and have achieved certain results. At present, the research on the golden jackal optimization algorithm at home and abroad can be divided into three parts: the improvement research on the golden jackal optimization algorithm itself, the algorithm research on the fusion of the golden jackal optimization algorithm and other optimization algorithms, and the engineering application of the golden jackal optimization algorithm in different fields [22].

With the continuous development of unmanned flight technology, unmanned aerial vehicles (UAVs) have attracted much attention due to their potential to work in complex and dangerous environments [23]. Path planning and design is an important part of the UAV mission system, which requires obtaining a safe, feasible, and smooth flight path from the starting position to the destination position under specific constraints [24,25]. Therefore, the problem of UAV path planning can be regarded as a complex optimization problem that requires effective algorithms to solve. For the problem of UAV path planning, researchers have proposed many methods to solve it, such as traditional methods such as the artificial potential field algorithm [26], Rapidly-exploring Random Tree (RRT) [27], and neural network algorithms [28], as well as emerging reinforcement learning algorithms such as the Q-learning algorithm [29]. However, these methods require a large amount of computing time and resources.

Swarm intelligence optimization algorithms are intelligent algorithms inspired by natural behaviors. As an effective method to solve complex optimization problems, more and more researchers apply meta-heuristic algorithms to solve the problem of UAV path planning. Liu et al. proposed three improved sparrow search algorithms, which enabled UAVs to obtain high-quality flight paths [30]. Zhu et al. proposed a cooperative evolution spider monkey optimization algorithm for the path planning and obstacle avoidance of unmanned combat vehicles [31]. These meta-heuristic algorithms have more advantages in solving path planning problems in complex environments. PSO algorithms and GWO algorithms are widely used to solve the problem of UAV path planning. Xu et al. proposes a Rotating Particle Swarm Optimization (RPSO) algorithm that rotates to search for targets in high-dimensional space and a new double-layer coding (DLC) model, which can always generate feasible trajectories in complex environments [32]. Liu et al. proposed an improved adaptive grey wolf optimization (AGWO) algorithm to solve the three-dimensional path planning problem of UAVs in complex environments [33]. Chen et al. proposed a hybrid SSA to implement the optimal 3D deployment of multi-UAV base stations, and the proposed method outperforms the traditional methods in terms of the sum log-rate utility and throughput [34].

Although algorithms such as the PSO algorithm, the GWO algorithm, and the sparrow search algorithm (SSA) can be used to solve the problem of UAV path planning, there are still problems such as slow convergence speed, poor stability, and it is easy to fall into local optima. To improve these problems, this paper proposes a golden jackal optimization algorithm combining the sine–cosine algorithm and the Cauchy mutation, referred to as the SCMGJO compound algorithm, for the path planning of UAVs in complex and dangerous environments. It mainly uses the introduction of the tent mapping reverse learning to initialize the population, and it applies the sine–cosine strategy to the update of the prey position in the algorithm to enhance the diversity of the population in the later iterations, avoid premature convergence, and improve the accuracy of optimization and the convergence speed; at the same time, the Cauchy mutation strategy is used to help the algorithm jump out of local optima and further accelerate the convergence. This smooths the flight path and generates a path that is more suitable for the flight of the UAV, thereby further saving the energy consumption of the UAV and the shortest flight path.

## 3. Golden Jackal Optimization Algorithm (GJO)

Golden jackals, medium-sized canids, are chiefly distributed in Africa and Asia. Their diet mainly consists of small mammals, birds, insects, and so on, and they typically employ hunting and predation to obtain food [35]. During foraging, golden jackals search for food resources within a certain area of their familiar territory. Golden jackals frequently forage in the form of golden jackal pairs, with male jackals doing the hunting while the female jackals follow, and they usually forage in groups. Multiple golden jackals will collaborate to hunt, increasing the success rate [36]. When foraging, golden jackals will adjust their search strategies based on the abundance of food and the location of other competitors to acquire more food resources [37]. The algorithm mainly comprises three stages: searching for prey and moving towards it; surrounding and stimulating the prey until it stops moving; and pouncing on the prey [38].

(1)Initialization phase

The mathematical formula for the initialization process of the golden jackal population is expressed as follows:(1)$$Y_{0}=Y_{min}+rand\left(Y_{max}-Y_{min}\right)$$

In the formula, $Y_{min}$ denotes the lower limit of $Y_{0}$, and $Y_{max}$ denotes the upper limit of $Y_{0}$, while *rand* denotes a uniformly distributed random value within $\left(0,1\right)$.

The initial prey matrix is as follows:(2)$$Prey=\left[\begin{array}{cccc} Y_{1,1} & Y_{1,2} & \cdots & Y_{1,d\;}\\ Y_{2,1} & Y_{2,2} & \cdots & Y_{2,d\;}\\ \vdots & \vdots & & \vdots \\ Y_{n,1} & Y_{n,2} & \cdots & Y_{n,d\;} \end{array}\right]$$

In the formula, *n* denotes the number of the population, and *d* denotes the dimension of the population.

(2)Search phase

Golden jackal pairs search for prey within their territory and gradually move towards the prey. During the process of searching for prey, the male golden jackal leads and the female golden jackal follows. The calculation method of the relative position between the golden jackal and the prey is as follows:(3)$$Y_{1}\left(t\right)=Y_{M}\left(t\right)-E*\left|Y_{M}\left(t\right)-rl*Prey\left(t\right)\right|$$
(4)$$Y_{2}\left(t\right)=Y_{F}\left(t\right)-E*\left|Y_{F}\left(t\right)-rl*Prey\left(t\right)\right|$$

In the formula, *t* denotes the current number of iterations, $Y_{M}(t)$ denotes the position of the male golden jackal after *t* iterations, $Y_{F}(t)$ denotes the position of the female golden jackal after *t* iterations, Prey(t) denotes the position vector of the prey, *rl* represents an arbitrary vector based on the Levy distribution, $Y_{1}(t)$ denotes the position of the male golden jackal corresponding to the prey after update, and $Y_{2}(t)$ denotes the position of the female golden jackal corresponding to the prey after update. *E* denotes the energy during the prey’s escape process, and the calculation formula is as follows:(5)$$E=E_{1}*E_{0}$$

In the formula, $E_{1}$ denotes the decrease in the energy of the prey, and $E_{0}$ denotes the initial energy of the prey. The calculation formula of $E_{0}$ is as follows:(6)$$E_{0}=2*r-1$$

In the formula, *r* denotes a uniformly distributed random value within $\;\left[0,1\right]$. The calculation formula of $E_{1}$ is as follows:(7)$$E_{1}=c_{1}*\left(1-\frac{t}{T}\right)$$

In the formula, $c_{1}$ is a constant, and $c_{1}=1.5$; *T* denotes the maximum number of iterations. During the iteration, $E_{1}$ decreases linearly from 1.5 to 0. The calculation formula of rl is as follows:(8)$$rl=0.05*Levy\left(y\right)$$

The Levy flight function is the following:(9)$$Levy\left(y\right)=0.01*\frac{\mu *\sigma }{v^{\frac{1}{\beta }}}$$

In the formula, $\sigma =\frac{T\left(1+\beta \right)sin{\left(\frac{\pi *\beta }{2}\right)}^{\frac{1}{\beta }}}{T\left(\frac{1+\beta }{2}\right)*2^{\frac{\beta -1}{2}}\;}\;$; β=1.5; μ and *v* both denote uniformly distributed random values in $\left[0,1\right]$; and the golden jackal update formula is calculated as the following:(10)$$Y\left(t+1\right)=\frac{Y_{1}\left(t\right)+Y_{2}\left(t\right)}{2}$$

(3)Siege phase

When the prey is disturbed by the golden jackal, its escape energy will gradually decrease, and then the golden jackal will tightly surround the prey. Once the prey is successfully surrounded by the golden jackal, they will launch an attack. In this stage, the position updates of the male and female golden jackals can be calculated as the following:(11)$$Y_{1}\left(t\right)=Y_{M}\left(t\right)-E*\left|rl*Y_{M}\left(t\right)-Prey\left(t\right)\right|$$
(12)$$Y_{2}\left(t\right)=Y_{F}\left(t\right)-E*\left|rl*Y_{F}\left(t\right)-Prey\left(t\right)\right|$$

In the formula, *t* denotes the current number of iterations, $Y_{M}(t)$ denotes the position of the male golden jackal after *t* iterations, $Y_{F}(t)$ denotes the position of the female golden jackal after *t* iterations, Prey(t) denotes the position of the prey after the *t* iteration, $Y_{1}(t)$ denotes the position corresponding to the prey after the male golden jackal is updated in the t iteration, and $Y_{2}(t)$ denotes the position corresponding to the prey after the male golden jackal is updated in the *t* iteration.

(4)Conversion of global search and local search

The parameter *E* denotes the energy of the prey’s escape. In the process of the golden jackal’s foraging, the condition for the golden jackal to switch from the prey search stage to the prey besieging stage is determined by E. When $\left|E\right|\geq 1$, the golden jackal conducts a global search to find the position of the prey; when $\left|E\right|<1$, the golden jackal conducts a local search to besiege the prey.

## 4. Sine–Cosine and Cauchy Mutation of Golden Jackal Optimization Algorithm (SCMGJO)

### 4.1. Tent Mapping Reverse Learning

Tent mapping, also known as the tent map, is a special form of mapping in the field of mathematics, with the characteristic of piecewise linearity. Its functional image is named after its unique tent shape and has a wide range of application values in multiple fields [39]. The core feature of this mapping lies in its piecewise linear property, which allows it to perform mapping operations according to different slopes in different numerical intervals, thus exhibiting a rich and diverse dynamic behavior.

In swarm intelligence optimization algorithms, population initialization is an extremely crucial step. It aims to provide sufficient initial conditions and high-quality search spaces for the optimization process of individuals, ensuring that the population distribution has a high density, thereby accelerating the optimization speed of the algorithm [40]. Because tent mapping has significant advantages over other mapping methods in terms of uniformity and ergodicity, this paper chooses to use tent mapping to initialize the golden jackal population. In addition, in order to further improve the quality of the initial population, this paper also introduces the reverse learning strategy. By screening and refining the initial population, more excellent golden jackal individuals are selected, thus providing a more favorable environmental condition for the optimization process of the algorithm and then improving the convergence speed of the algorithm.

The expression of the tent mapping is the following:(13)$$X_{n+1}=\left\{\begin{array}{l} 2X_{n},\;\;0\leq X_{n}\leq \frac{1}{2}\\ 2\left(1-X_{n}\right),\;\;\frac{1}{2}\leq X_{n}\leq 1 \end{array}\right.$$

Assuming that a set of feasible solutions in a D-dimensional golden jackal population is $\left\{X_{1},X_{2},X_{3},\cdots X_{D}\right\},X\in \left[lb,ub\right]$, then its reverse solution is $X\prime =\left\{X_{1}^{\prime },X_{2}^{\prime },X_{3}^{\prime },\cdots X_{D}\prime \right\},X\prime =lb+ub-X_{i}$. The total initialization of the golden jackal population with the reverse learning of the tent mapping is divided into three steps:

Step 1. Use the tent mapping to initialize the position of the golden jackal $x_{ij}(i=1,2\cdots D,j=1,2\cdots N)$, where *N* represents the population size.

Step 2. Generate reverse individual positions $x_{ij}$ based on the definition of the reverse solution, for the initial population position of the golden jackal.

Step 3. Sort the individual positions generated by the two methods according to the fitness level and select the one with the highest fitness as the male golden jackal and the one with the second highest fitness as the female golden jackal.

### 4.2. Sine–Cosine Algorithm

In the process of golden jackal hunting, the position of the prey is particularly crucial, which profoundly affects the forward trajectory of the entire golden jackal population [41]. However, due to the different positions of the prey, when the food searched by the male golden jackal happens to be at the local optimum, this may lead to the convergence of the population, that is, a large number of golden jackals tend to the same position. In this case, the golden jackal population will come to a standstill, the diversity of population positions will be compromised, and the risk of falling into local extreme values will increase [42].

To address this phenomenon, this paper introduces the sine–cosine algorithm (SCA) in the position update process of the golden jackal in the GJO algorithm. The position update formula of the SCA is as follows [43]:(14)$$X_{i}^{t+1}=\left\{\begin{array}{c} X_{i}^{t}+r_{1}*\sin\left(r2\right)*\left|r_{3}P_{i}^{t}-X_{i}^{t}\right|,r_{4}<0.5\\ X_{i}^{t}+r_{1}*\cos\left(r2\right)*\left|r_{3}P_{i}^{t}-X_{i}^{t}\right|,r_{4}\geq 0.5 \end{array}\right.$$

SCA can enhance the global search ability of the GJO algorithm and more effectively avoid falling into local optimal solutions. By using SCA to dynamically adjust the position of the male golden jackal, the search efficiency and accuracy of the algorithm can be improved while maintaining the diversity of the population. In the formula, $r_{2}$ denotes a uniformly distributed random value in $\left[0,2\pi \right]$, $r_{3}$ denotes a uniformly distributed random value in $\left[-2,2\right]$, and $r_{4}$ denotes a uniformly distributed random value in $\left[0,1\right]$. The calculation formula of $r_{1}$ is as follows [44]:(15)$$r_{1}=a-t\times \frac{a}{T}$$

In the formula, *a* is a constant, *t* denotes the number of iterations, and *T* denotes the maximum number of iterations.

When $\left|E\right|\geq 1$, a global search is required using Equations (16) and (17) to find the prey;
(16)$$Y_{M}^{\prime \left(t\right)}=\left\{\begin{array}{c} Y_{M}\left(t\right)+r_{1}*\sin\left(r2\right)*E*\left|r_{3}*rl*Prey\left(t\right)-Y_{M}\left(t\right)\right|,r_{4}<0.5\\ Y_{M}\left(t\right)+r_{1}*\cos\left(r2\right)*E*\left|r_{3}*rl*Prey\left(t\right)-Y_{M}\left(t\right)\right|,r_{4}\geq 0.5 \end{array}\right.$$
(17)$$Y_{F}^{\prime \left(t\right)}=\left\{\begin{array}{c} Y_{F}\left(t\right)+r_{1}*\sin\left(r2\right)*E*\left|r_{3}*rl*Prey\left(t\right)-Y_{F}\left(t\right)\right|,r_{4}<0.5\\ Y_{F}\left(t\right)+r_{1}*\cos\left(r2\right)*E*\left|r_{3}*rl*Prey\left(t\right)-Y_{F}\left(t\right)\right|,r_{4}\geq 0.5 \end{array}\right.$$

And when $\left|E\right|<1$, local search needs to be carried out using Equations (18) and (19) to encircle the prey.
(18)$$Y_{M}^{\prime \prime \left(t\right)}=\left\{\begin{array}{c} Y_{M}\left(t\right)+r_{1}*\sin\left(r2\right)*E*\left|r_{3}*Prey\left(t\right)-rl*Y_{M}\left(t\right)\right|,r_{4}<0.5\\ Y_{M}\left(t\right)+r_{1}*\cos\left(r2\right)*E*\left|r_{3}*Prey\left(t\right)-rl*Y_{M}\left(t\right)\right|,r_{4}\geq 0.5 \end{array}\right.$$
(19)$$Y_{F}^{\prime \prime \left(t\right)}=\left\{\begin{array}{c} Y_{F}\left(t\right)+r_{1}*\sin\left(r2\right)*E*\left|r_{3}*Prey\left(t\right)-rl*Y_{F}\left(t\right)\right|,r_{4}<0.5\\ Y_{F}\left(t\right)+r_{1}*\cos\left(r2\right)*E*\left|r_{3}*Prey\left(t\right)-rl*Y_{F}\left(t\right)\right|,r_{4}\geq 0.5 \end{array}\right.$$

### 4.3. Cauchy Mutation

A clear collaborative pattern is obtained from the foraging process of the golden jackals: in the search phase, it is mainly dominated by the male golden jackal with the best fitness in the current golden jackal population, while the female golden jackal plays the role of a follower, foraging closely around the optimal solution. In order to prevent the algorithm from falling into the predicament of a local optimum, we introduce the Cauchy mutation strategy into the position update formula of the golden jackal, aiming to enhance the global optimization ability of the algorithm. After the application of the Cauchy mutation strategy, the position update formula of the golden jackal is optimized, and the position update formula is the following:(20)$$Y_{M}\left(t\right)=Y_{1}\left(t\right)+Y_{1}\left(t\right)*Cauchy\left(0,1\right)$$

In the formula, $Cauchy\left(0,1\right)$ is the standard Cauchy distribution function; the one-dimensional Cauchy mutation function with the origin as the center is as follows:(21)$$f\left(x\right)=\frac{1}{\pi }\left(\frac{1}{x^{2}+1}\right),-\infty <x<\infty$$

The Cauchy distribution, as a continuous probability distribution, has a similar shape to the standard normal distribution. However, it has unique characteristics: the value at the origin is relatively small, while at the two ends of the distribution, it presents a flatter shape, which makes the rate of approaching zero of the Cauchy distribution slower. Due to this characteristic, the Cauchy distribution often has a more significant perturbation effect compared to the normal distribution.

Given this advantage of the Cauchy distribution, it is applied to the process of updating the position of the male golden jackal, and the individuals are perturbed using Cauchy mutation to expand the search scope of the SCMGJO. In this way, the algorithm can better explore the solution space, thereby enhancing its ability to jump out of local optimal solutions and further enhancing the global search performance of the algorithm.

### 4.4. Novel Update Rules

The flowchart of the SCMGJO algorithm is shown in Figure 1. At the same time, the algorithm flow includes eight steps.

Step 1: Initialize the position of the golden jackal population using the reverse learning of the tent mapping and initialize the position of the prey and the number of iterations.

Step 2: Calculate the fitness of each golden jackal individual.

Step 3: Select the optimal solution as the position information of the male golden jackal $Y_{M}(t)$ and the secondary optimal solution as the position information of the female golden jackal $Y_{F}(t)$.

Step 4: Calculate the escape energy E according to Equations (5)–(7).

Step 5: If $\left|E\right|\geq 1$, update the position using Equations (16), (17) and (20); otherwise $\left|E\right|<1$, update the position using Equations (18)–(20).

Step 6: Determine whether the maximum number of iterations has been reached. If the maximum number of iterations has not been reached, return to Step 2 to continue; otherwise, end the program.

## 5. Experiment Results and Analysis

### 5.1. Experimental Setup

In order to fully verify the effectiveness of the SCMGJO algorithm designed in this study in solving optimization problems, a series of experiments were conducted in this study. All algorithms were run on the experimental platform of Windows 11, 16GB memory, and a 64-bit system, and experiments were conducted using Matlab2022a. This paper selects the whale optimization algorithm (WOA) [45], ant colony optimization algorithm (ACO) [46], ant lion optimization algorithm (ALO) [47], gray wolf optimization algorithm (GWO) [48], golden jackal optimization algorithm (GJO), and the sine–cosine golden jackal optimization algorithm (SCGJO) [39] for comparison.

### 5.2. Function Testing and Performance Indicators

When conducting in-depth research on the CEC test function set, we selected 23 classical benchmark functions and conducted detailed simulation experimental analyses. These functions are not only used to comprehensively evaluate the global optimization performance of the algorithm but also to measure its convergence efficiency. Table 1 details the expressions, dimensions, domain, and theoretical optimal values of these benchmark functions. Among them, F1 to F13 are multidimensional test functions, while F14 to F23 are fixed-dimensional test functions.

In order to ensure the fairness of the comparison between different algorithms, we uniformly set the experimental parameters: the population size of the six algorithms is 50, and the maximum number of iterations is 500. For the multidimensional test functions F1 to F13, we set the dimensions to 10, 30, and 50, while F14 to F23 were tested under a fixed dimension. Each algorithm was run independently 100 times, and we selected the optimal value, average value, and standard deviation as the key evaluation indicators. The better the performance of the algorithm, the closer the optimal value is to the theoretical minimum value, and the smaller the average value and standard deviation.

After a large number of tests and data analysis, we obtained the data in Table 2, Table 3, Table 4 and Table 5. Among them, Table 2 presents the comparative data of each algorithm for F1 to F13 in 10 dimensions; Table 3 shows the comparison in 30 dimensions; Table 4 shows the data in 50 dimensions; Table 5 shows the data in 100 dimensions; and Table 6 reflects the comparative results of each algorithm for F14 to F23 under a fixed dimension. These data clearly reflect the excellent performance of the SCMGJO on the multidimensional test functions F1 to F13. Compared with the GJO, the SCMGJO, which fuses sine and cosine and Cauchy mutation, has significantly improved in solution accuracy, and the optimization performance is more stable. Compared with other algorithms, the SCMGJO shows a significant advantage in the optimization effect.

In addition, we also obtained the convergence curves of 100 independent runs. Figure 2 shows the 23 convergence curves of functions F1 to F13 in 30 dimensions and F14 to F23 in a fixed dimension. For the test functions F1, F2, F3, F4, F5, F7, F9, F10, F11, F15, F16, and F19, the optimization results of the SCMGJO algorithm reached the theoretical optimal value, which further verifies the effectiveness of the sine–cosine strategy and Cauchy mutation in escaping local optima and rapid searching and also indicates that the SCMGJO algorithm has strong optimization ability and convergence speed.

To sum up, through this detailed simulation experiment analysis, we can conclude that the SCMGJO demonstrates excellent global optimization ability and convergence speed on multidimensional test functions, especially in the outstanding performance on single-peak benchmark functions. At the same time, its stability and robustness are also superior to other comparative algorithms, providing strong support and reference for solving practical optimization problems.

### 5.3. Tension Compression Spring Design Problem

In order to further verify the ability of the SCMGJO algorithm in practical engineering applications, the tension compression spring design problem is introduced. The tension compression spring design problem is to select the design variables that minimize the mass of the spring while satisfying a set of given constraints, where the design variables include the wire diameter d$\left(x_{1}\right)$, the average diameter of the spring coils D$\left(x_{2}\right)$, and the number of effective coils of the spring P$\left(x_{3}\right)$. Figure 3 shows the structural diagram of the tension compression spring, and the cost function of this problem is as follows:(22)$$minf\left(x\right)=\left(x_{3}+2\right)x_{2}x_{1}^{2}$$

The constraint conditions are as follows:(23)$$g_{1}\left(x\right)=1-\frac{x_{3}x_{2}^{3}}{71785x_{1}^{4}}\leq 0$$
(24)$$g_{2}\left(x\right)=\frac{4x_{2}^{2}-x_{1}x_{2}}{12566\left(x_{2}x_{1}^{3}-x_{1}^{4}\right)}+\frac{1}{5108x_{1}^{2}}-1\leq 0$$
(25)$$g_{3}\left(x\right)=1-\frac{140.45x_{1}}{x_{3}x_{2}^{2}}\leq 0$$
(26)$$g_{4}\left(x\right)=\frac{x_{1}+x_{2}}{1.5}-1\leq 0$$

The boundary constraint conditions are shown as follows:(27)$$\left\{\begin{array}{c} 0.05\leq x_{1}\leq 2\\ 0.25\leq x_{2}\leq 1.3\\ 2\leq x_{3}\leq 15 \end{array}\right.$$

We apply the SCMGJO proposed in this paper to the design of tension compression springs and conduct comparative simulation experiments with five other algorithms, setting the number of iterations to 100 and 300, respectively.

Figure 4 and Figure 5 show the 100-iteration and 300-iteration diagrams of the comparison of different algorithms in solving the tension/compression spring design problem, respectively. Table 7 and Table 8 show the objective function values of different algorithms in solving the tension/compression spring design problem under 100 iterations and 300 iterations, respectively. From Table 6 and Table 7, it can be seen that the objective function value obtained by the SCMGJO is the smallest, which is better than the objective function values of the WOA, ACO, ALO, GWO, and GJO, indicating that the improved SCMGJO can minimize the manufacturing cost of this project compared to other algorithms, overcome multiple interference factors, and demonstrate the superior ability of the SCMGJO in solving the tension/compression spring design problem in this paper.

### 5.4. Three-Bar Truss Design Problem

The core objective of the three-bar truss design problem is to minimize the volume of the entire three-bar truss by adjusting the cross-sectional areas $x_{1}$ and $x_{2}$. During this process, each truss component is subject to specific stress σ limitations. This optimization problem not only involves a nonlinear fitness function but also three nonlinear inequality constraints, and it also needs to handle two continuous decision variables. Figure 6 shows the structural diagram of the three-bar truss, and the specific form of the objective function is as follows:(28)$$minf\left(x\right)=\left(2\sqrt{2}x_{1}+x_{2}\right)\cdot L$$

The constraint conditions are as follows:(29)$$g_{1}\left(x\right)=\frac{2\sqrt{2}x_{1}+x_{2}}{\sqrt{2}x_{1}^{2}+2x_{1}x_{2}}P-\sigma \leq 0$$
(30)$$g_{2}\left(x\right)=\frac{x_{2}}{\sqrt{2}x_{1}^{2}+2x_{1}x_{2}}P-\sigma \leq 0$$
(31)$$g_{3}\left(x\right)=\frac{1}{x_{1}+\sqrt{2}x_{2}}P-\sigma \leq 0$$

The boundary constraint conditions are shown as follows:(32)$$\left\{\begin{array}{c} 0\leq x_{1}\leq 1\\ 0\leq x_{2}\leq 1 \end{array}\right.$$

The SCMGJO proposed in this paper is applied to the design of the three-bar truss design problem, and comparative simulation experiments are carried out with the other five algorithms, with the iteration times set to 100 times and 500 times, respectively.

Figure 7 and Figure 8 show the 100-iteration and 500-iteration diagrams of the comparison of different algorithms in solving the three-bar truss design problem, respectively. Table 9 and Table 10 show the objective function values of different algorithms in solving the three-bar truss design problem under 100 iterations and 500 iterations, respectively. From Table 8 and Table 9, it can be seen that the objective function value obtained by the SCMGJO is the smallest in both 100 iterations and 300 iterations, which is better than the objective function values of the WOA, ACO, ALO, GWO, and GJO, indicating that the improved SCMGJO can minimize the volume of the three-bar truss in this project compared to other algorithms, effectively avoiding local optima, and demonstrating the superior ability of the SCMGJO in solving the three-bar truss design problem in this project.

### 5.5. Unmanned Aerial Vehicle (UAV) Path Planning

When applying the SCMGJO to handle the path planning problem of unmanned aerial vehicles, the fitness value of an individual is calculated through the cost function. The cost function is jointly determined by the path length cost, vertical height cost, smoothing processing cost, and safety cost, and the optimal solution is found in each iteration process to achieve the selection of the route with the minimum flight path cost. When solving the path planning with the SCMGJO, the cost function and constraint conditions of the unmanned vehicle path planning problem are first determined, and the SCMGJO is initialized; then, a certain number of individuals are generated for the population using the tent chaos mapping, the fitness value of the individuals in the population is calculated, and the optimal solution and the suboptimal solution are selected as the positions of the male and female golden leopards of the SCMGJO for the iterative search of the SCMGJO. In each iteration, the position is updated using the sine–cosine strategy and the Cauchy mutation strategy until the algorithm termination condition is met. The lowest cost map of the output path and the path planning map of the unmanned machine are output.

To verify the effectiveness of the SCMGJO proposed in this paper in the unmanned aerial vehicle path planning problem, this experiment sets up two different scenarios for simulation experiments. The flight space size is 1000 m × 1000 m × 400 m, the number of selected populations is 100, and the number of iterations is 10. The whale optimization algorithm (WOA), ant colony optimization algorithm (ACO), ant lion optimization algorithm (ALO), gray wolf optimization algorithm (GWO), golden jaguar optimization algorithm (GJO), and the sine–cosine and Cauchy mutation golden jackal optimization algorithm (SCMGJO) are selected for comparison. The unmanned aerial vehicle needs to fly from the starting point to the end point and bypass the threat source. The cost function is used to measure the performance of the algorithm in the unmanned aerial vehicle path planning. The lower the cost, the better the performance. In this experiment, two different scenarios are used in the simulation to test the performance of each algorithm in the unmanned aerial vehicle path planning.

Scene 1: There are six threat sources in this scene, and Table 11 shows the threat source information. The starting position of the unmanned vehicle is (200, 100, 150), and the ending position is (800, 800, 150). Figure 9 shows the three-dimensional scene after setting the parameters.

In Scene 1, the top views and three-dimensional simulations of the unmanned aerial vehicle path planning for each algorithm are shown in Figure 10 and Figure 11, respectively. From these figures, it can be seen that SCMGJO outperforms other algorithms in both top views and three-dimensional coordinates. The GJO does not avoid the last threat source and flies straight through it, while SCMGJO has a significant performance improvement compared to the GJO. The WOA’s planned path can be clearly seen in both the top view and three-dimensional view, which is inferior to SCMGJO. Although the GWO does not fly through the threat source like the GJO, it has a detour behavior. The ALO and the ACO are similar to the SCMGJO in the top view, but in the three-dimensional coordinates, it can be clearly seen that the ALO and the ACO have multiple fluctuations in the vertical height, which affects the overall cost. Figure 12 shows the iterative cost curve of each algorithm.

Scene 2: There are nine threat sources in this scene, and Table 12 shows the threat source information. The starting position of the unmanned vehicle is (200, 100, 150), and the ending position is (800, 800, 150). Figure 13 shows the three-dimensional scene after setting the parameters.

In Scene 2, the top views and three-dimensional simulations of the unmanned aerial vehicle path planning for each algorithm are shown in Figure 14 and Figure 15, respectively. From these figures, it can be seen that when the number of threat sources is increased, making Scene 2 more complex than Scene 1, the SCMGJO algorithm still maintains superior performance. The traditional GJO algorithm makes the same mistake as in Scene 1, failing to avoid the threat sources and flying straight through them. The GWO algorithm makes the same mistake as the GJO algorithm, indicating that the performance of the GWO algorithm needs to be improved in this scenario. The WOA and the ACO algorithm take a detour when passing through the last threat source, increasing the length of the path and affecting the overall cost. The ALO algorithm shows the same phenomenon as in Scene 1, with multiple fluctuations in the vertical height, increasing the cost of path planning. Figure 16 shows the iterative cost curve of each algorithm in Scene 2.

To sum up, the improved SCMGJO addresses the problems of slow convergence speed, insufficient accuracy, weakened optimization ability in the later stage, and the tendency to fall into local extremum that exist in the traditional GJO. This unmanned aerial vehicle path planning simulation experiment verifies that the SCMGJO algorithm has the ability to perform path planning in both simple and complex environments and has better performance and a shorter execution time.

## 6. Conclusions

This study introduces a sine–cosine and Cauchy mutation golden jackal optimization algorithm (SCMGJO), which combines sine–cosine and Cauchy mutations to overcome the shortcomings of the GJO algorithm. In the algorithm design, the tent mapping reverse learning strategy is used to initialize the golden leopard population. The introduction of sine–cosine and Cauchy mutations significantly improves the global and local search capabilities of the algorithm. To verify the performance of the SCMGJO algorithm, 23 benchmark test functions were selected and in-depth simulation experiments were conducted. The experimental results show that compared with the traditional GJO algorithm, SCMGJO has significant advantages in solution accuracy and convergence speed. In addition, compared with other algorithms, SCMGJO shows a significant improvement in the optimization capabilities of multidimensional single-peak and multi-peak functions. Furthermore, engineering application simulations are introduced. The simulation results show that SCMGJO outperforms the traditional GJO and other compared algorithms in terms of solution accuracy and convergence speed. From a large amount of experimental data, it can be concluded that SCMGJO has significantly improved the optimization capabilities of multidimensional functions, but it still requires continuous optimization for fixed-dimensional functions.

Looking to the future, we will continue to conduct in-depth research to optimize the SCMGJO algorithm, enhance the algorithm’s optimization capabilities in fixed low-dimensional spaces, and expand the application of the SCMGJO algorithm in more engineering fields. We will continuously improve the algorithm’s universality and robustness to better solve various practical optimization problems. Through continuous exploration and improvement, the SCMGJO algorithm will play a greater role in the field of optimization.

## Figures and Tables

**Figure 1 biomimetics-09-00270-f001:**
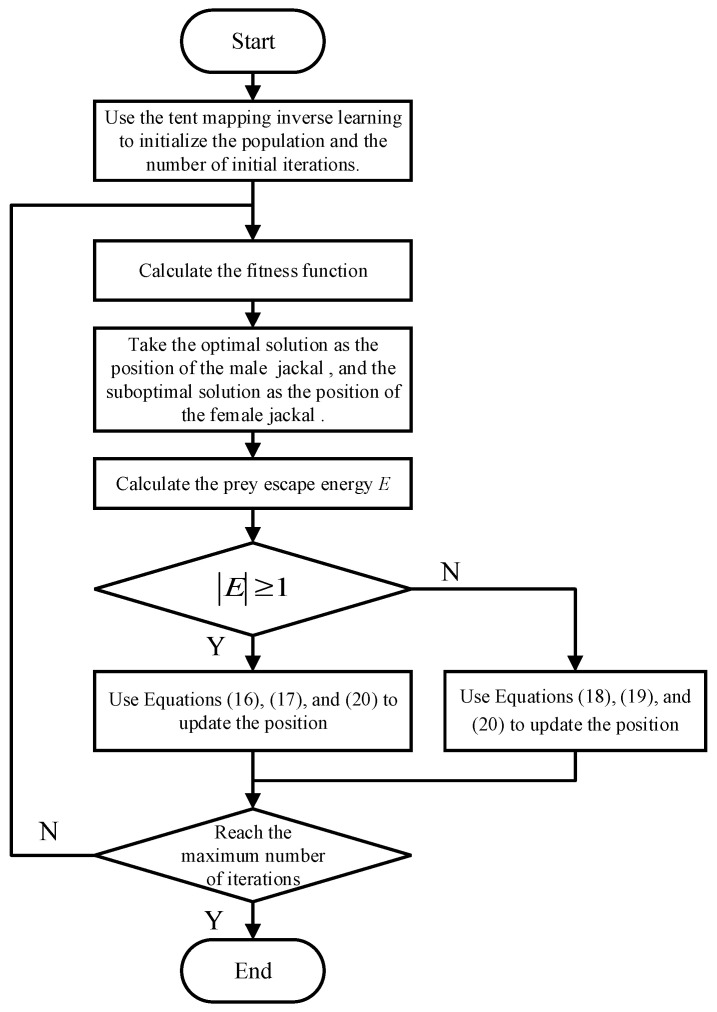
The flowchart of the SCMGJO algorithm.

**Figure 2 biomimetics-09-00270-f002:**
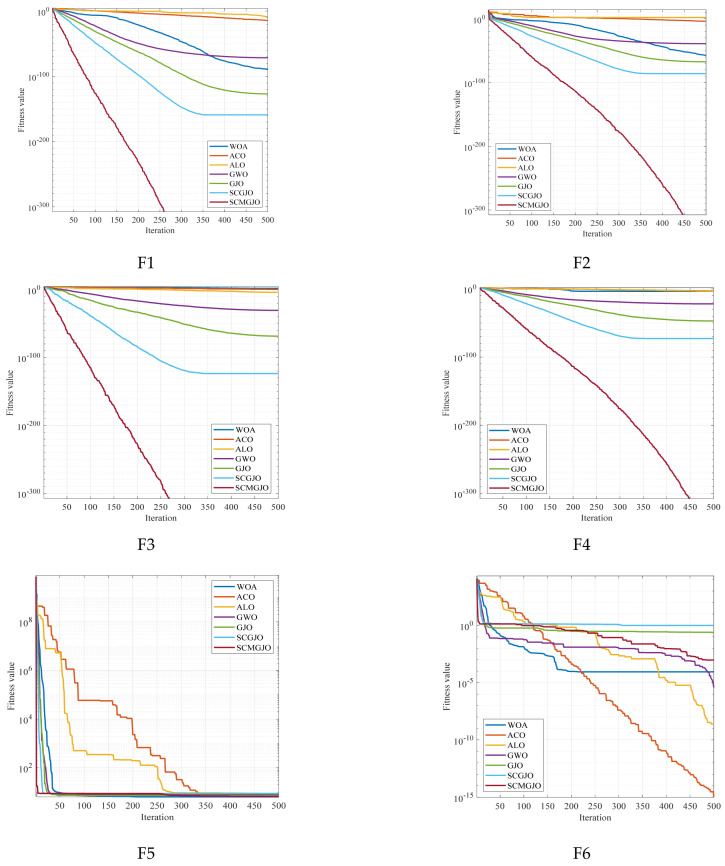
Iteration curve diagrams (Dim = 30 and fixed-dimensional).

**Figure 3 biomimetics-09-00270-f003:**
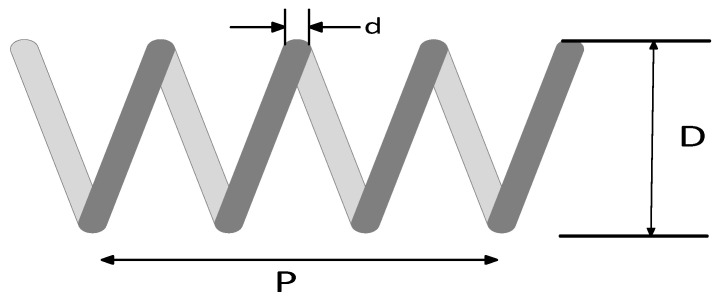
The structural diagram of the tension compression spring.

**Figure 4 biomimetics-09-00270-f004:**
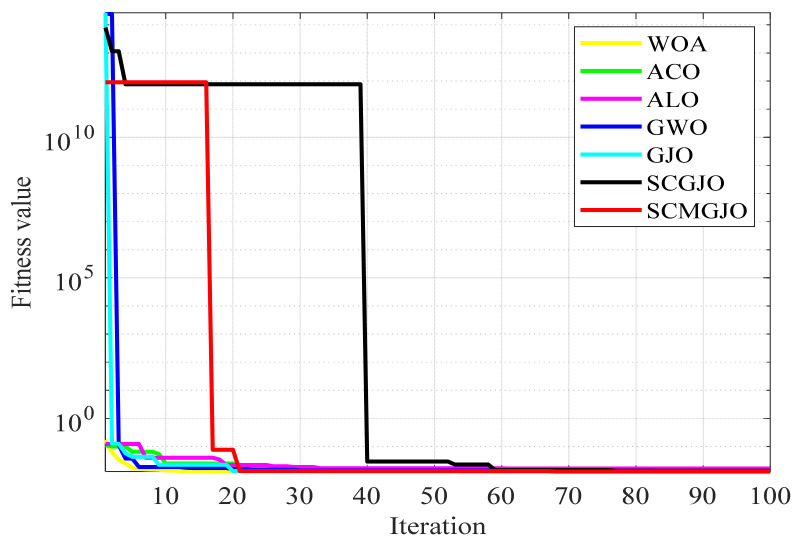
Iterative curves for the design problem of tension compression springs (100 iterations).

**Figure 5 biomimetics-09-00270-f005:**
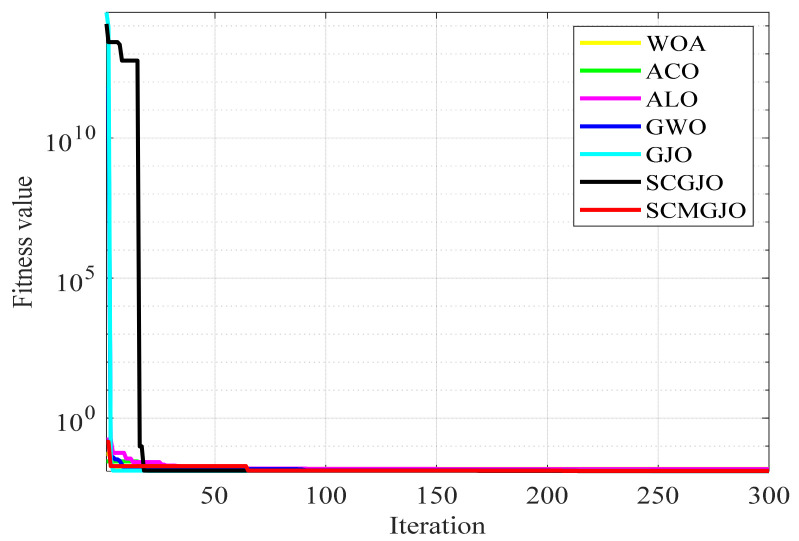
Iterative curves for the design problem of tension compression springs (300 iterations).

**Figure 6 biomimetics-09-00270-f006:**
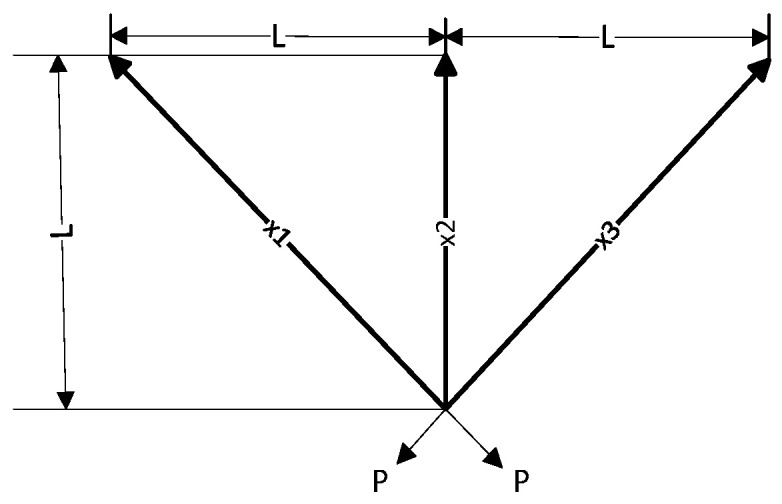
The structural diagram of the three-bar truss.

**Figure 7 biomimetics-09-00270-f007:**
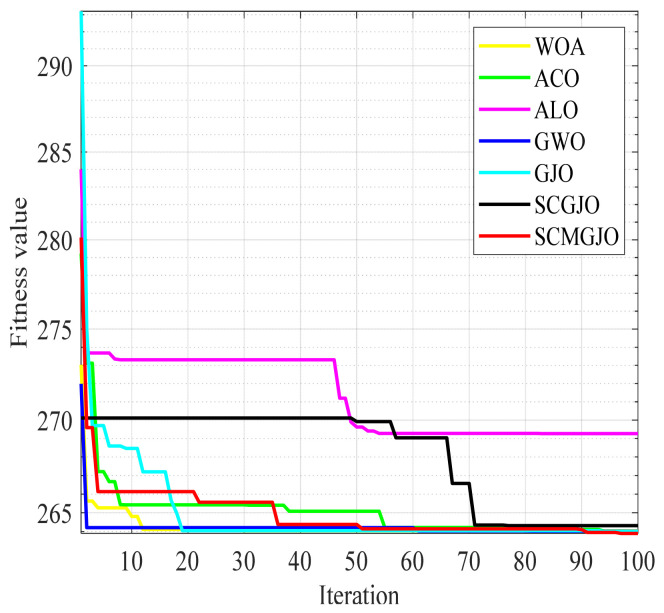
Iteration curve for three-bar truss design problem (100 iterations).

**Figure 8 biomimetics-09-00270-f008:**
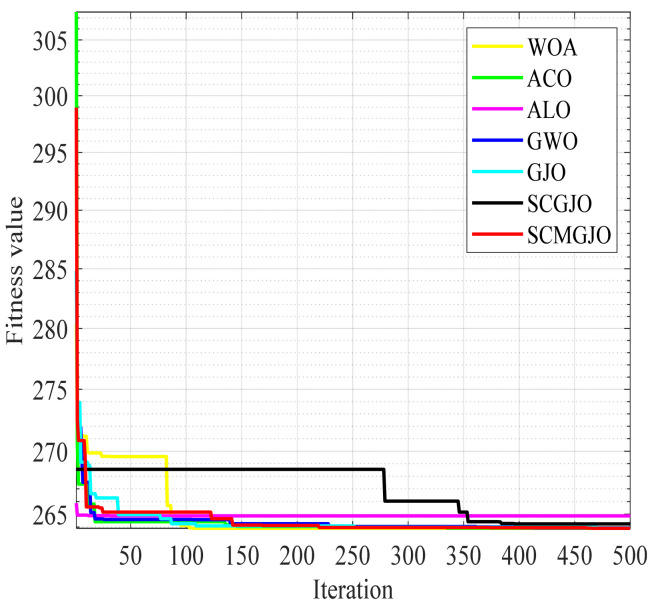
Iteration curve for three-bar truss design problem (500 iterations).

**Figure 9 biomimetics-09-00270-f009:**
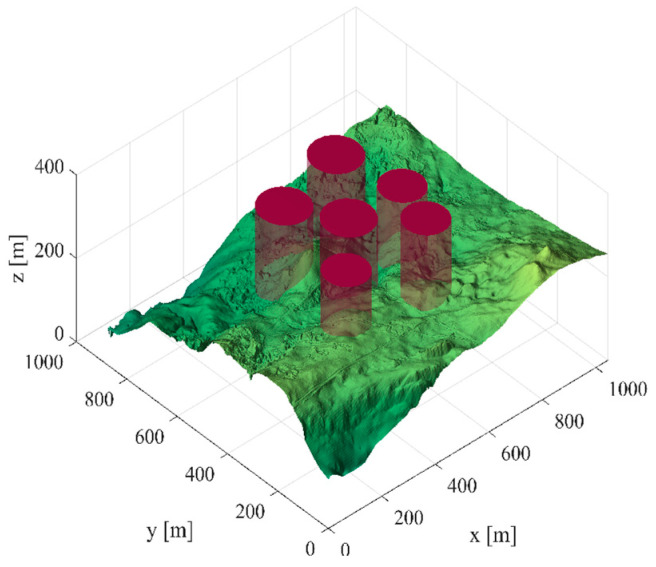
The 3D scene graph (6 threat sources).

**Figure 10 biomimetics-09-00270-f010:**
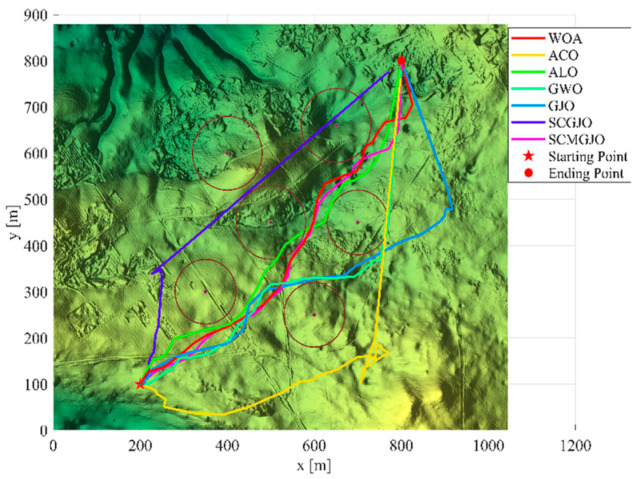
Top view of the drone’s path (6 threat sources).

**Figure 11 biomimetics-09-00270-f011:**
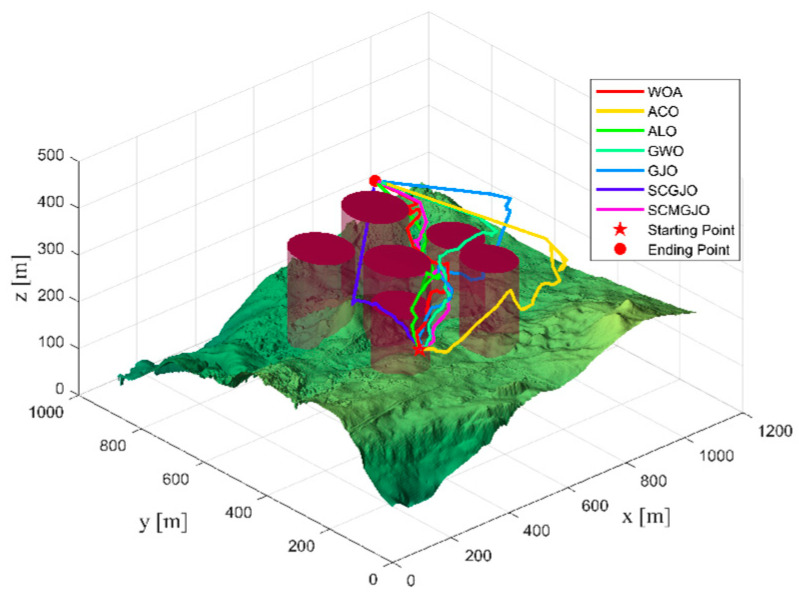
The 3D diagram of the drone’s path (6 threat sources).

**Figure 12 biomimetics-09-00270-f012:**
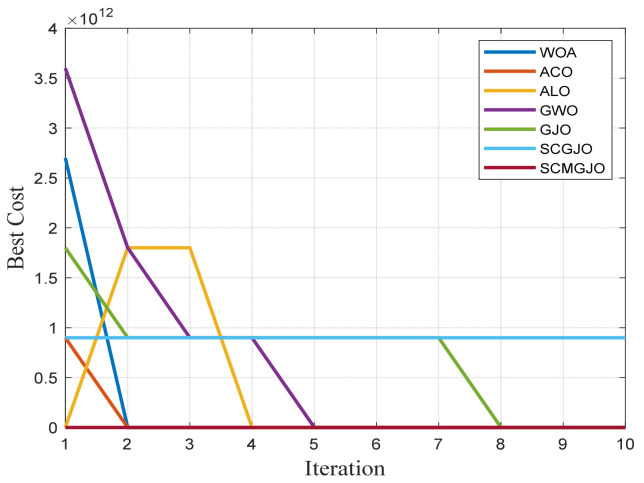
Iterative curve of the drone (6 threat sources).

**Figure 13 biomimetics-09-00270-f013:**
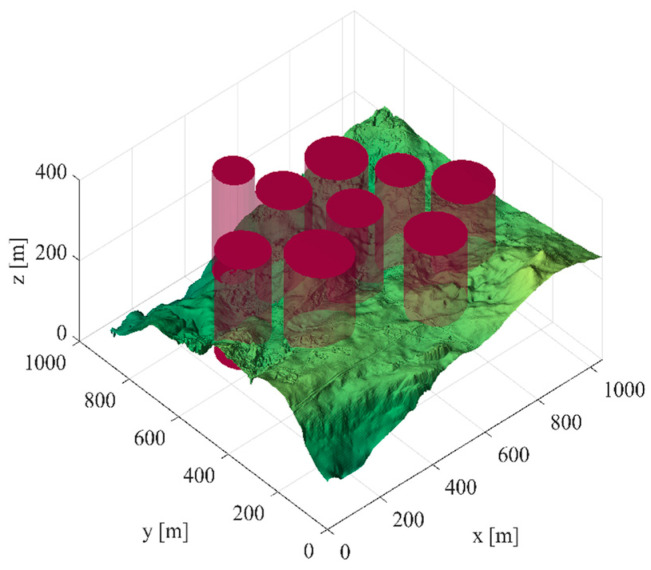
The 3D scene graph (9 threat sources).

**Figure 14 biomimetics-09-00270-f014:**
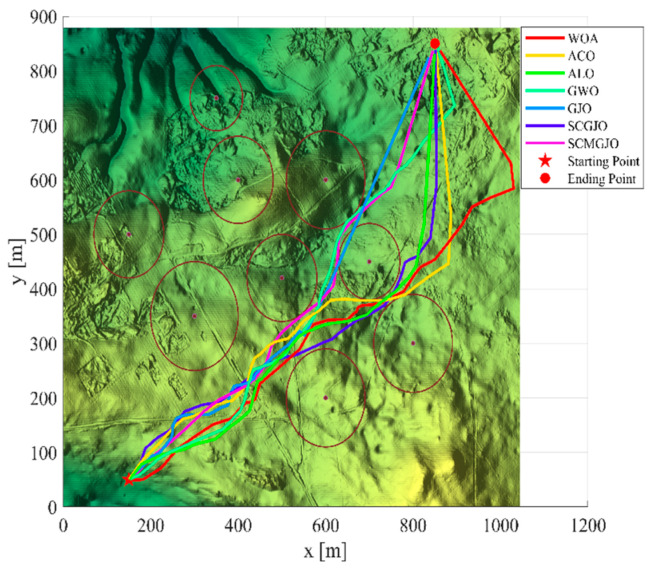
Top view of the drone’s path (9 threat sources).

**Figure 15 biomimetics-09-00270-f015:**
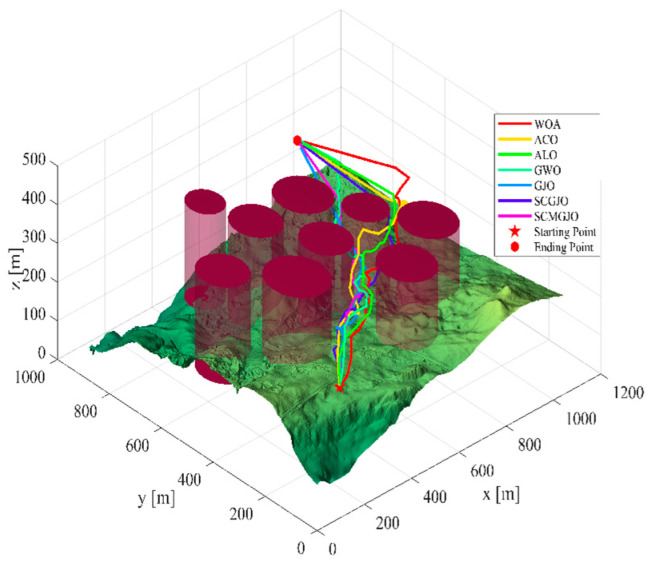
The 3D diagram of the drone’s path (9 threat sources).

**Figure 16 biomimetics-09-00270-f016:**
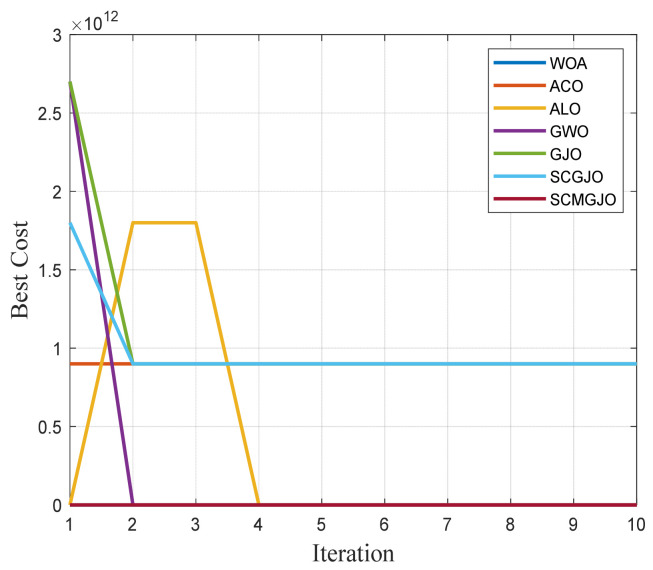
Iterative curve of the drone(9 threat sources).

**Table 1 biomimetics-09-00270-t001:** The benchmark functions.

Function	Name	Dim	Range	Best
F1	Sphere	10/30/50	[−100, 100]	0
F2	Schwefel 2.22	10/30/50	[−10, 10]	0
F3	Schwefel 1.2	10/30/50	[−100, 100]	0
F4	Schwefel 2.21	10/30/50	[−100, 100]	0
F5	Rosenbrock	10/30/50	[−30, 30]	0
F6	Step	10/30/50	[−100, 100]	0
F7	Quartic with noise	10/30/50	[−1.28, 1.28]	0
F8	Schwefel 2.26	10/30/50	[−500, 500]	−12,569.5
F9	Rastrigin	10/30/50	[−5.12, 5.12]	0
F10	Ackley	10/30/50	[−32, 32]	0
F11	Griewank	10/30/50	[−600, 600]	0
F12	Penalized1	10/30/50	[−50, 50]	0
F13	Penalized2	10/30/50	[−50, 50]	0
F14	Shekel’s Foxholes	2	[−65.536, 65.536]	1
F15	Kowalik	4	[−5, 5]	0.000308
F16	Six-Hump Camel Back	2	[−5, 5]	−1.0316
F17	Branin	2	[−5, 10], [0, 15]	0.398
F18	Goldstein-Price	2	[−2, 2]	3
F19	Hartman’s Family 1	3	[0, 1]	−3.86
F20	Hartman’s Family2	6	[0, 1]	−3.32
F21	Shekel’s Family1	4	[0, 10]	−10.15
F22	Shekel’s Family2	4	[0, 10]	−10.4
F23	Shekel’s Family3	4	[0, 10]	−10.536

**Table 2 biomimetics-09-00270-t002:** The optimal fitness of each algorithm (Dim = 10).

Function	Item	WOA	ACO	ALO	GWO	GJO	SCGJO	SCMGJO
F1	min	2.67 × 10^−87^	3.34 × 10^−15^	2.98 × 10^−9^	3.76 × 10^−73^	5.58 × 10^−129^	1.41 × 10^−160^	0.00 × 10^0^
	mean	−6.67 × 10^−45^	5.58 × 10^−9^	−1.57 × 10^−7^	3.85 × 10^−38^	−5.07 × 10^−66^	1.53 × 10^−81^	5.70 × 10^−164^
	std	1.57 × 10^−44^	1.83 × 10^−8^	1.82 × 10^−5^	2.00 × 10^−37^	2.43 × 10^−65^	3.62 × 10^−81^	2.71 × 10^−1^
F2	min	8.94 × 10^−61^	4.34 × 10^−7^	1.75 × 10^2^	2.39 × 10^−39^	1.53 × 10^−69^	1.49 × 10^−83^	0.00 × 10^0^
	mean	−8.71 × 10^−62^	1.91 × 10^−8^	1.43 × 10^1^	−5.38 × 10^−41^	−7.22 × 10^−71^	−7.91 × 10^−85^	0.00 × 10^0^
	std	1.03 × 10^−61^	4.79 × 10^−8^	3.20 × 10^1^	2.47 × 10^−40^	1.44 × 10^−70^	1.75 × 10^−84^	3.06 × 10^−1^
F3	min	1.00 × 10^2^	3.50 × 10^0^	1.09 × 10^−5^	2.58 × 10^−33^	1.82 × 10^−73^	9.56 × 10^−122^	0.00 × 10^0^
	mean	3.38 × 10^−1^	−9.77 × 10^−2^	−1.31 × 10^−4^	−1.55 × 10^−18^	−1.30 × 10^−38^	−1.41 × 10^−62^	−1.49 × 10^−163^
	std	5.87 × 10^0^	8.06 × 10^−1^	1.89 × 10^−3^	2.96 × 10^−17^	2.31 × 10^−37^	1.52 × 10^−61^	3.36 × 10^−1^
F4	min	2.03 × 10^−3^	7.44 × 10^−4^	4.29 × 10^−4^	1.40 × 10^−24^	1.18 × 10^−48^	4.13 × 10^−73^	0.00 × 10^0^
	mean	−2.65 × 10^−5^	−1.60 × 10^−4^	−5.92 × 10^−5^	2.79 × 10^−25^	−4.67 × 10^−49^	−3.33 × 10^−74^	0.00 × 10^0^
	std	1.44 × 10^−3^	4.52 × 10^−4^	4.01 × 10^−4^	1.44 × 10^−24^	1.14 × 10^−48^	3.26 × 10^−73^	2.82 × 10^−1^
F5	min	6.89 × 10^0^	6.62 × 10^0^	8.14 × 10^0^	7.18 × 10^0^	8.05 × 10^0^	8.09 × 10^0^	8.10 × 10^0^
	mean	1.80 × 10^−1^	1.68 × 10^−1^	5.48 × 10^−2^	1.37 × 10^−1^	6.83 × 10^−2^	6.49 × 10^−2^	6.43 × 10^−2^
	std	2.63 × 10^−1^	2.64 × 10^−1^	1.03 × 10^−1^	2.42 × 10^−1^	1.46 × 10^−1^	1.47 × 10^−1^	3.45 × 10^−1^
F6	min	3.31 × 10^−5^	8.49 × 10^−15^	5.01 × 10^−9^	3.17 × 10^−6^	2.62 × 10^−5^	7.54 × 10^−1^	4.77 × 10^−1^
	mean	−5.00 × 10^−1^	−5.00 × 10^−1^	−5.00 × 10^−1^	−5.00 × 10^−1^	−5.00 × 10^−1^	−3.41 × 10^−1^	−4.50 × 10^−1^
	std	1.92 × 10^−3^	3.04 × 10^−8^	2.34 × 10^−5^	5.46 × 10^−4^	1.70 × 10^−3^	2.36 × 10^−1^	3.22 × 10^−1^
F7	min	6.11 × 10^−4^	6.19 × 10^−3^	4.68 × 10^−3^	5.34 × 10^−4^	3.82 × 10^−4^	2.85 × 10^−5^	1.02 × 10^−5^
	mean	7.47 × 10^−3^	−1.23 × 10^−2^	2.64 × 10^−2^	1.09 × 10^−2^	−2.91 × 10^−3^	5.35 × 10^−4^	1.44 × 10^−4^
	std	5.05 × 10^−2^	6.75 × 10^−2^	7.67 × 10^−2^	4.77 × 10^−2^	3.77 × 10^−2^	6.16 × 10^−3^	2.99 × 10^−1^
F8	min	−6.36 × 10^2^	−5.09 × 10^2^	−5.44 × 10^2^	−5.48 × 10^2^	−4.01 × 10^2^	−3.91 × 10^2^	−5.57 × 10^2^
	mean	6.55 × 10^1^	−6.85 × 10^1^	−1.00 × 10^2^	3.07 × 10^1^	2.07 × 10^1^	−3.08 × 10^1^	−6.61 × 10^1^
	std	1.18 × 10^−2^	2.17 × 10^1^	0.00 × 10^0^	5.96 × 10^1^	5.57 × 10^1^	6.74 × 10^1^	2.88 × 10^−1^
F9	min	0.00 × 10^0^	3.71 × 10^1^	1.69 × 10^1^	0.00 × 10^0^	0.00 × 10^0^	0.00 × 10^0^	0.00 × 10^0^
	mean	−2.08 × 10^−12^	5.90 × 10^−2^	9.95 × 10^−2^	−7.17 × 10^−10^	−1.42 × 10^−9^	1.97 × 10^−10^	9.77 × 10^−10^
	std	2.72 × 10^−9^	9.91 × 10^−1^	1.36 × 10^0^	3.47 × 10^−9^	2.00 × 10^−9^	2.78 × 10^−9^	2.55 × 10^−1^
F10	min	4.00 × 10^−15^	2.05 × 10^1^	1.16 × 10^0^	2.02 × 10^1^	4.00 × 10^−15^	4.44 × 10^−16^	4.44 × 10^−16^
	mean	−3.06 × 10^−17^	8.13 × 10^1^	−8.72 × 10^−2^	−1.11 × 10^0^	3.40 × 10^−16^	−9.87 × 10^−17^	−3.79 × 10^−17^
	std	1.14 × 10^−15^	4.87 × 10^2^	2.76 × 10^−1^	7.85 × 10^1^	1.23 × 10^−15^	2.70 × 10^−16^	2.83 × 10^−1^
F11	min	0.00 × 10^0^	5.27 × 10^−1^	4.78 × 10^−1^	0.00 × 10^0^	0.00 × 10^0^	0.00 × 10^0^	0.00 × 10^0^
	mean	−7.13 × 10^−9^	3.66 × 10^−1^	1.62 × 10^0^	−1.11 × 10^−11^	4.96 × 10^−9^	−2.96 × 10^−10^	1.12 × 10^−9^
	std	6.74 × 10^−9^	7.58 × 10^0^	1.45 × 10^1^	1.29 × 10^−8^	1.32 × 10^−8^	1.57 × 10^−8^	2.84 × 10^−1^
F12	min	1.02 × 10^−4^	5.42 × 10^−15^	1.79 × 10^−9^	9.97 × 10^−7^	5.75 × 10^−6^	1.97 × 10^−1^	5.89 × 10^−2^
	mean	−9.98 × 10^−1^	−1.00 × 10^0^	−1.00 × 10^0^	−1.00 × 10^0^	−1.00 × 10^0^	−5.05 × 10^−1^	−5.93 × 10^−1^
	std	2.39 × 10^−2^	1.49 × 10^−7^	1.00 × 10^−4^	2.12 × 10^−3^	5.09 × 10^−3^	5.32 × 10^−1^	4.06 × 10^−1^
F13	min	6.14 × 10^−4^	4.12 × 10^−14^	8.88 × 10^−9^	2.41 × 10^−6^	9.76 × 10^−2^	5.35 × 10^−1^	8.01 × 10^−1^
	mean	9.96 × 10^−1^	1.00 × 10^0^	1.00 × 10^0^	1.00 × 10^0^	9.04 × 10^−1^	5.10 × 10^−1^	4.99 × 10^−1^
	std	2.54 × 10^−2^	1.78 × 10^−7^	9.60 × 10^−5^	1.63 × 10^−3^	3.05 × 10^−1^	5.64 × 10^−1^	4.22 × 10^−1^

**Table 3 biomimetics-09-00270-t003:** The optimal fitness of each algorithm (Dim = 30).

Function	Item	WOA	ACO	ALO	GWO	GJO	SCGJO	SCMGJO
F1	min	2.17 × 10^−86^	4.06 × 10^3^	2.17 × 10^−4^	1.34 × 10^−33^	4.32 × 10^−63^	7.17 × 10^−111^	0.00 × 10^0^
	mean	−9.99 × 10^−46^	4.67 × 10^−1^	1.47 × 10^−5^	1.94 × 10^−18^	2.07 × 10^−33^	−8.80 × 10^−57^	−2.44 × 10^−164^
	std	2.73 × 10^−44^	1.18 × 10^1^	2.74 × 10^−3^	6.52 × 10^−18^	1.20 × 10^−32^	1.29 × 10^−56^	0.00 × 10^0^
F2	min	1.75 × 10^−57^	7.58 × 10^3^	6.77 × 10^2^	8.64 × 10^−19^	4.15 × 10^−36^	1.22 × 10^−55^	0.00 × 10^0^
	mean	2.66 × 10^−59^	6.74 × 10^0^	1.76 × 10^1^	6.33 × 10^−21^	−8.66 × 10^−39^	5.25 × 10^−58^	0.00 × 10^0^
	std	1.10 × 10^−58^	1.03 × 10^2^	3.28 × 10^1^	2.89 × 10^−20^	1.45 × 10^−37^	5.16 × 10^−57^	0.00 × 10^0^
F3	min	1.98 × 10^4^	1.38 × 10^5^	2.08 × 10^3^	2.56 × 10^−10^	3.37 × 10^−24^	1.52 × 10^−78^	0.00 × 10^0^
	mean	5.77 × 10^−1^	1.89 × 10^0^	−1.90 × 10^−1^	−6.53 × 10^−8^	−1.17 × 10^−14^	−7.13 × 10^−43^	−1.95 × 10^−164^
	std	2.93 × 10^1^	7.82 × 10^1^	1.48 × 10^1^	4.99 × 10^−6^	5.26 × 10^−13^	3.35 × 10^−40^	0.00 × 10^0^
F4	min	1.56 × 10^1^	8.56 × 10^1^	1.46 × 10^1^	5.45 × 10^−9^	1.04 × 10^−18^	2.65 × 10^−46^	0.00 × 10^0^
	mean	−1.15 × 10^0^	−7.37 × 10^0^	−7.28 × 10^−2^	1.37 × 10^−9^	−8.97 × 10^−20^	6.19 × 10^−48^	0.00 × 10^0^
	std	9.63 × 10^0^	5.23 × 10^1^	1.23 × 10^1^	5.20 × 10^−9^	9.39 × 10^−19^	1.67 × 10^−46^	0.00 × 10^0^
F5	min	2.78 × 10^1^	2.01 × 10^3^	3.85 × 10^3^	2.79 × 10^1^	2.62 × 10^1^	2.81 × 10^1^	2.89 × 10^1^
	mean	3.24 × 10^−2^	−3.09 × 10^1^	6.08 × 10^−1^	2.70 × 10^−2^	7.39 × 10^−2^	2.20 × 10^−2^	3.06 × 10^−3^
	std	8.34 × 10^−2^	1.82 × 10^2^	1.29 × 10^0^	8.62 × 10^−2^	2.08 × 10^−1^	8.96 × 10^−2^	4.68 × 10^−3^
F6	min	6.84 × 10^−2^	4.62 × 10^3^	2.46 × 10^−4^	3.76 × 10^−5^	2.26 × 10^0^	5.02 × 10^0^	4.74 × 10^0^
	mean	−4.95 × 10^−1^	−1.85 × 10^0^	−5.00 × 10^−1^	−5.00 × 10^−1^	−3.50 × 10^−1^	−1.77 × 10^−1^	−2.00 × 10^−1^
	std	4.83 × 10^−2^	1.25 × 10^1^	2.91 × 10^−3^	1.13 × 10^−3^	2.33 × 10^−1^	2.54 × 10^−1^	2.50 × 10^−1^
F7	min	6.73 × 10^−4^	4.68 × 10^−1^	1.31 × 10^−1^	5.65 × 10^−4^	1.78 × 10^−4^	3.19 × 10^−5^	2.03 × 10^−5^
	mean	1.27 × 10^−3^	7.05 × 10^−1^	−1.07 × 10^−2^	−9.84 × 10^−4^	−3.86 × 10^−5^	−7.60 × 10^−5^	−1.13 × 10^−4^
	std	3.07 × 10^−2^	2.72 × 10^−1^	1.48 × 10^−1^	2.21 × 10^−2^	1.73 × 10^−2^	1.22 × 10^−3^	2.85 × 10^−4^
F8	min	−1.87 × 10^3^	−1.83 × 10^3^	−1.63 × 10^3^	−1.17 × 10^3^	−9.83 × 10^2^	−8.19 × 10^2^	−1.53 × 10^3^
	mean	6.25 × 10^1^	−6.16 × 10^1^	−1.00 × 10^2^	1.12 × 10^1^	5.44 × 10^0^	−9.29 × 10^0^	5.09 × 10^1^
	std	1.67 × 10^1^	3.38 × 10^1^	0.00 × 10^0^	5.73 × 10^1^	5.45 × 10^1^	5.90 × 10^1^	3.68 × 10^1^
F9	min	0.00 × 10^0^	3.25 × 10^1^	9.65 × 10^1^	5.68 × 10^−14^	0.00 × 10^0^	0.00 × 10^0^	0.00 × 10^0^
	mean	−2.33 × 10^−9^	−2.18 × 10^0^	−3.32 × 10^−2^	1.26 × 10^−9^	−2.52 × 10^−10^	5.14 × 10^−10^	9.02 × 10^−11^
	std	2.47 × 10^−9^	9.82 × 10^0^	1.82 × 10^0^	5.89 × 10^−9^	3.62 × 10^−9^	3.14 × 10^−9^	2.65 × 10^−10^
F10	min	4.44 × 10^−16^	2.10 × 10^1^	2.81 × 10^0^	2.09 × 10^1^	2.08 × 10^1^	4.00 × 10^−15^	4.44 × 10^−16^
	mean	−5.81 × 10^−17^	6.10 × 10^1^	1.28 × 10^−1^	9.32 × 10^0^	6.12 × 10^0^	−3.01 × 10^−16^	4.98 × 10^−18^
	std	1.21 × 10^−16^	8.24 × 10^1^	7.42 × 10^−1^	5.79 × 10^1^	6.46 × 10^1^	1.25 × 10^−15^	1.31 × 10^−16^
F11	min	0.00 × 10^0^	1.89 × 10^0^	1.07 × 10^−2^	1.14 × 10^−2^	0.00 × 10^0^	0.00 × 10^0^	0.00 × 10^0^
	mean	−2.25 × 10^−9^	−1.88 × 10^0^	2.84 × 10^−1^	−2.87 × 10^−1^	−1.14 × 10^−9^	1.27 × 10^−9^	8.92 × 10^−10^
	std	1.49 × 10^−8^	1.09 × 10^1^	1.13 × 10^0^	1.13 × 10^0^	1.61 × 10^−8^	7.64 × 10^−9^	3.39 × 10^−9^
F12	min	2.78 × 10^−2^	1.20 × 10^−2^	1.17 × 10^1^	6.89 × 10^−6^	3.00 × 10^−1^	7.42 × 10^−1^	6.32 × 10^−1^
	mean	−8.51 × 10^−1^	1.45 × 10^0^	−8.05 × 10^−1^	−1.00 × 10^0^	−4.93 × 10^−1^	−2.47 × 10^−1^	−3.62 × 10^−1^
	std	3.22 × 10^−1^	2.74 × 10^1^	7.28 × 10^0^	5.91 × 10^−3^	5.07 × 10^−1^	4.47 × 10^−1^	4.81 × 10^−1^
F13	min	2.28 × 10^−1^	3.49 × 10^−1^	7.75 × 10^0^	6.30 × 10^−1^	1.64 × 10^0^	2.57 × 10^0^	2.90 × 10^0^
	mean	9.44 × 10^−1^	7.66 × 10^0^	1.06 × 10^0^	7.76 × 10^−1^	4.50 × 10^−1^	1.87 × 10^−1^	5.72 × 10^−2^
	std	2.55 × 10^−1^	2.86 × 10^1^	1.62 × 10^0^	4.07 × 10^−1^	4.97 × 10^−1^	4.29 × 10^−1^	2.16 × 10^−1^

**Table 4 biomimetics-09-00270-t004:** The optimal fitness of each algorithm (Dim = 50).

Function	Item	WOA	ACO	ALO	GWO	GJO	SCGJO	SCMGJO
F1	min	9.21 × 10^−93^	1.12 × 10^−5^	4.62 × 10^−2^	8.92 × 10^−24^	1.35 × 10^−45^	4.33 × 10^−89^	0.00 × 10^0^
	mean	1.68 × 10^−48^	−1.12 × 10^1^	−5.18 × 10^−4^	−2.45 × 10^−14^	−1.67 × 10^−24^	−1.62 × 10^−47^	6.06 × 10^−165^
	std	1.36 × 10^−47^	4.65 × 10^1^	3.07 × 10^−2^	4.26 × 10^−13^	4.98 × 10^−24^	9.40 × 10^−46^	0.00 × 10^0^
F2	min	5.46 × 10^−52^	1.55 × 10^2^	9.75 × 10^2^	2.55 × 10^−13^	5.42 × 10^−27^	5.37 × 10^−49^	0.00 × 10^0^
	mean	1.09 × 10^−53^	1.57 × 10^1^	1.73 × 10^1^	−7.99 × 10^−16^	3.60 × 10^−29^	8.17 × 10^−52^	0.00 × 10^0^
	std	5.26 × 10^−53^	7.68 × 10^1^	3.21 × 10^1^	5.17 × 10^−15^	1.09 × 10^−28^	1.39 × 10^−50^	0.00 × 10^0^
F3	min	1.38 × 10^5^	3.86 × 10^5^	7.56 × 10^3^	8.00 × 10^−3^	5.97 × 10^−11^	3.79 × 10^−67^	0.00 × 10^0^
	mean	−1.11 × 10^0^	7.50 × 10^−1^	−7.97 × 10^−3^	1.84 × 10^−4^	−2.43 × 10^−8^	−7.99 × 10^−37^	−1.49 × 10^−165^
	std	3.43 × 10^1^	5.46 × 10^1^	2.03 × 10^1^	1.98 × 10^−2^	1.54 × 10^−6^	1.10 × 10^−34^	0.00 × 10^0^
F4	min	7.71 × 10^−1^	9.08 × 10^1^	2.63 × 10^1^	2.28 × 10^−5^	1.15 × 10^−11^	7.25 × 10^−36^	0.00 × 10^0^
	mean	−1.49 × 10^−1^	6.20 × 10^−1^	−5.90 × 10^−1^	−1.61 × 10^−6^	−2.22 × 10^−12^	1.08 × 10^−36^	0.00 × 10^0^
	std	3.24 × 10^−1^	5.80 × 10^1^	1.95 × 10^1^	2.07 × 10^−5^	9.06 × 10^−12^	3.36 × 10^−36^	0.00 × 10^0^
F5	min	4.84 × 10^1^	6.03 × 10^10^	1.34 × 10^3^	4.93 × 10^1^	4.63 × 10^1^	4.89 × 10^1^	4.87 × 10^1^
	mean	1.19 × 10^−2^	8.20 × 10^0^	2.95 × 10^−1^	1.20 × 10^−2^	4.45 × 10^−2^	2.56 × 10^−3^	5.16 × 10^−3^
	std	2.11 × 10^−2^	5.06 × 10^1^	1.12 × 10^0^	1.57 × 10^−1^	1.62 × 10^−1^	4.61 × 10^−3^	2.29 × 10^−2^
F6	min	2.88 × 10^−1^	1.26 × 10^5^	8.88 × 10^−2^	2.51 × 10^0^	6.70 × 10^0^	1.00 × 10^1^	8.78 × 10^0^
	mean	−4.88 × 10^−1^	−7.08 × 10^0^	−4.99 × 10^−1^	−3.99 × 10^−1^	−2.30 × 10^−1^	−9.77 × 10^−2^	−1.48 × 10^−1^
	std	7.58 × 10^−2^	5.03 × 10^1^	4.26 × 10^−2^	2.02 × 10^−1^	2.50 × 10^−1^	1.98 × 10^−01^	2.30 × 10^−1^
F7	min	2.86 × 10^−3^	1.18 × 10^3^	4.07 × 10^2^	1.73 × 10^−3^	2.64 × 10^−4^	4.25 × 10^−5^	1.81 × 10^−4^
	mean	−6.91 × 10^−3^	−8.43 × 10^−1^	4.39 × 10^−4^	−2.37 × 10^−3^	−1.55 × 10^−3^	3.81 × 10^−4^	−1.58 × 10^−4^
	std	3.12 × 10^−2^	4.70 × 10^1^	8.93 × 10^−1^	2.82 × 10^−2^	1.35 × 10^−2^	4.49 × 10^−3^	8.75 × 10^−4^
F8	min	−3.18 × 10^3^	−6.59 × 10^3^	−2.72 × 10^3^	−1.68 × 10^3^	−1.16 × 10^3^	−6.98 × 10^2^	−2.39 × 10^3^
	mean	6.55 × 10^1^	1.57 × 10^2^	−1.00 × 10^2^	8.97 × 10^0^	−9.61 × 10^0^	−7.22 × 10^0^	5.18 × 10^1^
	std	7.53 × 10^−2^	1.11 × 10^1^	0.00 × 10^0^	5.16 × 10^1^	4.89 × 10^1^	5.38 × 10^1^	3.78 × 10^1^
F9	min	0.00 × 10^0^	9.80 × 10^4^	3.48 × 10^2^	2.17 × 10^0^	0.00 × 10^0^	0.00 × 10^0^	0.00 × 10^0^
	mean	−5.84 × 10^−10^	6.28 × 10^0^	2.99 × 10^−1^	−3.86 × 10^−2^	−5.42 × 10^−10^	5.62 × 10^−10^	4.69 × 10^−11^
	std	3.07 × 10^−9^	4.42 × 10^1^	2.63 × 10^0^	1.97 × 10^−1^	3.62 × 10^−9^	3.01 × 10^−9^	6.70 × 10^−10^
F10	min	7.55 × 10^−15^	2.11 × 10^1^	4.44 × 10^−16^	2.11 × 10^1^	2.07 × 10^1^	4.00 × 10^−15^	4.44 × 10^−16^
	mean	7.69 × 10^−17^	9.50 × 10^1^	0.00 × 10^0^	1.31 × 10^1^	−7.46 × 10^−2^	3.77 × 10^−16^	5.08 × 10^−18^
	std	2.24 × 10^−15^	1.23 × 10^3^	0.00 × 10^0^	6.34 × 10^1^	3.55 × 10^1^	1.30 × 10^−15^	4.47 × 10^−17^
F11	min	0.00 × 10^0^	2.95 × 10^1^	1.71 × 10^−1^	0.00 × 10^0^	0.00 × 10^0^	0.00 × 10^0^	0.00 × 10^0^
	mean	2.40 × 10^−10^	−7.16 × 10^0^	3.50 × 10^−2^	−2.05 × 10^−9^	−2.63 × 10^−9^	1.76 × 10^−9^	−3.33 × 10^−11^
	std	8.27 × 10^−9^	4.77 × 10^1^	1.13 × 10^0^	2.95 × 10^−8^	2.28 × 10^−8^	8.27 × 10^−9^	3.75 × 10^−9^
F12	min	1.84 × 10^−2^	3.28 × 10^1^	4.18 × 10^1^	5.73 × 10^−2^	3.69 × 10^−1^	8.52 × 10^−1^	7.44 × 10^−1^
	mean	−9.12 × 10^−1^	3.61 × 10^0^	−3.26 × 10^−1^	−6.99 × 10^−1^	−4.53 × 10^−1^	−1.65 × 10^−1^	−2.04 × 10^−1^
	std	2.87 × 10^−1^	4.97 × 10^1^	8.85 × 10^0^	4.53 × 10^−1^	4.90 × 10^−1^	3.79 × 10^−1^	4.08 × 10^−1^
F13	min	2.26 × 10^−1^	4.40 × 10^1^	1.41 × 10^2^	1.30 × 10^0^	3.45 × 10^0^	4.47 × 10^0^	4.80 × 10^0^
	mean	9.70 × 10^−1^	6.18 × 10^0^	1.18 × 10^−2^	7.17 × 10^−1^	3.04 × 10^−1^	1.29 × 10^−1^	4.04 × 10^−2^
	std	1.99 × 10^−1^	4.89 × 10^1^	4.75 × 10^0^	4.26 × 10^−1^	4.46 × 10^−1^	3.48 × 10^−1^	1.98 × 10^−1^

**Table 5 biomimetics-09-00270-t005:** The optimal fitness of each algorithm (Dim = 100).

Function	Item	WOA	ACO	ALO	GWO	GJO	SCGJO	SCMGJO
F1	min	4.80 × 10^−87^	7.33 × 10^2^	8.38 × 10^2^	1.19 × 10^−15^	9.01 × 10^−32^	3.81 × 10^−76^	0.00 × 10^0^
	mean	−3.35 × 10^−46^	−6.82 × 10^−4^	−1.04 × 10^−2^	−4.28 × 10^−10^	−2.60 × 10^−18^	7.97 × 10^−41^	−2.05 × 10^−164^
	std	6.95 × 10^−45^	2.72 × 10^0^	2.91 × 10^0^	3.44 × 10^−9^	3.00 × 10^−17^	1.96 × 10^−39^	0.00 × 10^0^
F2	min	7.70 × 10^−54^	4.13 × 10^−3^	2.13 × 10^−106^	1.64 × 10^−8^	8.38 × 10^−19^	1.32 × 10^−40^	0.00 × 10^0^
	mean	6.14 × 10^−56^	−7.10 × 10^0^	1.41 × 10^0^	2.80 × 10^−12^	−1.25 × 10^−22^	−1.60 × 10^−43^	0.00 × 10^0^
	std	2.31 × 10^−55^	5.05 × 10^1^	4.14 × 10^1^	1.69 × 10^−10^	9.15 × 10^−21^	1.74 × 10^−42^	0.00 × 10^0^
F3	min	9.97 × 10^5^	3.11 × 10^4^	6.21 × 10^4^	6.70 × 10^0^	5.93 × 10^−6^	6.86 × 10^−60^	0.00 × 10^0^
	mean	1.22 × 10^0^	−2.60 × 10^−1^	−4.93 × 10^−1^	−6.05 × 10^−4^	−1.24 × 10^−6^	−5.35 × 10^−33^	−5.41 × 10^−165^
	std	4.97 × 10^1^	2.31 × 10^1^	2.89 × 10^1^	3.12 × 10^−1^	3.19 × 10^−4^	2.20 × 10^−31^	0.00 × 10^0^
F4	min	9.06 × 10^1^	2.87 × 10^1^	2.79 × 10^1^	1.01 × 10^−1^	1.75 × 10^−3^	1.84 × 10^−22^	0.00 × 10^0^
	mean	−1.24 × 10^0^	−3.46 × 10^0^	3.52 × 10^−1^	9.72 × 10^−3^	7.33 × 10^−5^	1.39 × 10^−23^	0.00 × 10^0^
	std	5.26 × 10^1^	1.81 × 10^1^	1.96 × 10^1^	7.73 × 10^−2^	1.21 × 10^−3^	7.75 × 10^−23^	0.00 × 10^0^
F5	min	9.73 × 10^1^	7.47 × 10^3^	1.11 × 10^3^	9.76 × 10^1^	9.86 × 10^1^	9.81 × 10^1^	9.81 × 10^1^
	mean	1.54 × 10^−2^	6.62 × 10^−1^	3.64 × 10^−1^	1.07 × 10^−2^	3.87 × 10^−3^	6.38 × 10^−3^	1.99 × 10^−3^
	std	5.11 × 10^−2^	6.18 × 10^0^	5.55 × 10^0^	4.80 × 10^−2^	4.89 × 10^−3^	4.84 × 10^−2^	4.01 × 10^−3^
F6	min	2.93 × 10^0^	7.63 × 10^2^	2.20 × 10^3^	8.42 × 10^0^	1.66 × 10^1^	2.18 × 10^1^	1.94 × 10^1^
	mean	−4.41 × 10^−1^	−4.94 × 10^−1^	−5.21 × 10^−1^	−3.31 × 10^−1^	−1.66 × 10^−1^	−6.83 × 10^−2^	−9.60 × 10^−2^
	std	1.62 × 10^−1^	2.78 × 10^0^	4.72 × 10^0^	2.37 × 10^−1^	2.34 × 10^−1^	1.78 × 10^−1^	1.99 × 10^−1^
F7	min	7.38 × 10^−4^	2.06 × 10^6^	4.33 × 10^6^	6.02 × 10^−3^	1.19 × 10^−3^	4.02 × 10^−5^	7.92 × 10^−5^
	mean	−4.88 × 10^−3^	−8.64 × 10^−2^	3.31 × 10^−1^	5.03 × 10^−4^	1.02 × 10^−3^	−5.53 × 10^−5^	1.22 × 10^−6^
	std	1.65 × 10^−2^	5.27 × 10^0^	6.25 × 10^0^	2.79 × 10^−2^	1.27 × 10^−2^	6.10 × 10^−4^	1.90 × 10^−5^
F8	min	−6.36 × 10^3^	−5.44 × 10^3^	−5.44 × 10^3^	−2.89 × 10^3^	−2.26 × 10^3^	−1.17 × 10^3^	−3.56 × 10^3^
	mean	6.55 × 10^1^	−1.00 × 10^2^	−1.00 × 10^2^	4.11 × 10^0^	6.50 × 10^−2^	−2.25 × 10^0^	3.72 × 10^1^
	std	4.19 × 10^−1^	0.00 × 10^0^	0.00 × 10^0^	4.78 × 10^1^	4.76 × 10^1^	7.16 × 10^1^	5.39 × 10^1^
F9	min	0.00 × 10^0^	4.30 × 10^3^	2.92 × 10^3^	4.29 × 10^0^	0.00 × 10^0^	0.00 × 10^0^	0.00 × 10^0^
	mean	−9.59 × 10^−10^	−2.09 × 10^−1^	−3.60 × 10^−1^	9.63 × 10^−3^	−1.38 × 10^−11^	2.99 × 10^−11^	−1.43 × 10^−10^
	std	1.77 × 10^−9^	6.52 × 10^0^	5.32 × 10^0^	1.73 × 10^−1^	3.25 × 10^−9^	2.53 × 10^−9^	7.83 × 10^−10^
F10	min	4.44 × 10^−16^	7.11 × 10^0^	4.44 × 10^−16^	2.13 × 10^1^	2.12 × 10^1^	4.00 × 10^−15^	4.44 × 10^−16^
	mean	5.08 × 10^−17^	3.53 × 10^−1^	0.00 × 10^0^	5.04 × 10^−2^	8.10 × 10^0^	−6.96 × 10^−17^	9.77 × 10^−19^
	std	8.73 × 10^−17^	2.05 × 10^0^	0.00 × 10^0^	5.49 × 10^1^	5.49 × 10^1^	1.37 × 10^−15^	1.20 × 10^−17^
F11	min	0.00 × 10^0^	1.24 × 10^0^	1.16 × 10^0^	6.66 × 10^−16^	0.00 × 10^0^	0.00 × 10^0^	0.00 × 10^0^
	mean	−8.22 × 10^−10^	2.44 × 10^−2^	−1.76 × 10^−3^	−3.44 × 10^−9^	−2.70 × 10^−10^	−2.22 × 10^−9^	5.50 × 10^−10^
	std	1.04 × 10^−8^	3.09 × 10^0^	2.56 × 10^0^	5.09 × 10^−8^	2.49 × 10^−8^	1.50 × 10^−8^	5.65 × 10^−9^
F12	min	2.78 × 10^−2^	6.55 × 10^3^	1.63 × 10^5^	1.93 × 10^−1^	6.21 × 10^−1^	1.03 × 10^0^	7.75 × 10^−1^
	mean	−9.17 × 10^−1^	1.85 × 10^−2^	−7.93 × 10^−1^	−5.65 × 10^−1^	−3.74 × 10^−1^	−1.55 × 10^−1^	−1.22 × 10^−1^
	std	3.19 × 10^−1^	9.44 × 10^0^	1.04 × 10^1^	4.92 × 10^−1^	4.73 × 10^−1^	3.77 × 10^−1^	3.29 × 10^−1^
F13	min	1.56 × 10^0^	2.78 × 10^5^	3.59 × 10^5^	6.05 × 10^0^	8.06 × 10^0^	9.70 × 10^0^	9.70 × 10^0^
	mean	8.66 × 10^−1^	−1.11 × 10^−1^	−1.56 × 10^−1^	3.72 × 10^−1^	1.91 × 10^−1^	3.00 × 10^−2^	2.98 × 10^−2^
	std	3.39 × 10^−1^	6.58 × 10^0^	6.77 × 10^0^	4.47 × 10^−1^	3.72 × 10^−1^	1.71 × 10^−1^	1.70 × 10^−1^

**Table 6 biomimetics-09-00270-t006:** The optimal fitness of each algorithm (fixed-dimensional).

Function	Item	WOA	ACO	ALO	GWO	GJO	SCGJO	SCMGJO
F14	min	9.98 × 10^−1^	9.98 × 10^−1^	9.98 × 10^−1^	9.98 × 10^−1^	9.98 × 10^−1^	9.98 × 10^−1^	9.98 × 10^−1^
	mean	−3.20 × 10^1^	−3.20 × 10^1^	−3.20 × 10^1^	−3.20 × 10^1^	−3.20 × 10^1^	−3.20 × 10^1^	−3.20 × 10^1^
	std	8.60 × 10^−3^	1.26 × 10^−2^	1.33 × 10^−2^	2.22 × 10^−2^	1.55 × 10^−2^	1.28 × 10^−1^	4.69 × 10^−3^
F15	min	9.19 × 10^−4^	1.23 × 10^−3^	1.04 × 10^−3^	9.21 × 10^−4^	1.19 × 10^−3^	5.21 × 10^−4^	9.79 × 10^−4^
	mean	1.22 × 10^1^	−2.25 × 10^3^	−3.90 × 10^1^	1.28 × 10^1^	−1.52 × 10^1^	9.29 × 10^−2^	1.54 × 10^−1^
	std	1.82 × 10^1^	2.91 × 10^3^	4.18 × 10^1^	1.94 × 10^1^	5.47 × 10^1^	9.21 × 10^−2^	3.42 × 10^−2^
F16	min	−1.03 × 10^0^	−1.03 × 10^0^	−1.03 × 10^0^	−1.03 × 10^0^	−1.03 × 10^0^	−1.03 × 10^0^	−1.03 × 10^0^
	mean	3.11 × 10^−1^	−3.11 × 10^−1^	−3.11 × 10^−1^	3.11 × 10^−1^	3.11 × 10^−1^	−3.12 × 10^−1^	3.13 × 10^−1^
	std	5.67 × 10^−1^	5.67 × 10^−1^	5.67 × 10^−1^	5.67 × 10^−1^	5.67 × 10^−1^	5.66 × 10^−1^	5.62 × 10^−1^
F17	min	3.98 × 10^−1^	4.04 × 10^−1^	3.98 × 10^−1^	3.98 × 10^−1^	3.98 × 10^−1^	3.98 × 10^−1^	3.99 × 10^−1^
	mean	2.77 × 10^1^	5.94 × 10^0^	1.15 × 10^1^	4.57 × 10^0^	4.57 × 10^0^	2.72 × 10^0^	2.71 × 10^0^
	std	8.12 × 10^0^	4.96 × 10^0^	2.96 × 10^1^	1.09 × 10^1^	1.09 × 10^1^	6.10 × 10^−1^	6.00 × 10^−1^
F18	min	3.00 × 10^0^	3.00 × 10^0^	3.00 × 10^0^	3.00 × 10^0^	3.00 × 10^0^	3.00 × 10^0^	3.00 × 10^0^
	mean	−5.01 × 10^−1^	−5.00 × 10^−1^	−5.00 × 10^−1^	−5.00 × 10^−1^	−5.00 × 10^−1^	−5.00 × 10^−1^	−5.00 × 10^−1^
	std	7.07 × 10^−1^	7.07 × 10^−1^	7.07 × 10^−1^	7.07 × 10^−1^	7.07 × 10^−1^	7.07 × 10^−1^	7.07 × 10^−1^
F19	min	−3.86 × 10^0^	−3.86 × 10^0^	−3.86 × 10^0^	−3.85 × 10^0^	−3.86 × 10^0^	−3.85 × 10^0^	−3.86 × 10^0^
	mean	5.07 × 10^−1^	5.08 × 10^−1^	5.08 × 10^−1^	4.67 × 10^−1^	5.00 × 10^−1^	4.67 × 10^−1^	4.69 × 10^−1^
	std	3.73 × 10^−1^	3.71 × 10^−1^	3.71 × 10^−1^	4.37 × 10^−1^	3.84 × 10^−1^	4.32 × 10^−1^	4.33 × 10^−1^
F20	min	0.00 × 10^0^	0.00 × 10^0^	0.00 × 10^0^	−3.07 × 10^0^	−3.32 × 10^0^	−2.64 × 10^0^	−1.92 × 10^0^
	mean	1.26 × 10^1^	2.06 × 10^1^	−2.02 × 10^1^	3.31 × 10^−1^	3.45 × 10^−1^	2.85 × 10^−1^	2.34 × 10^−1^
	std	5.92 × 10^1^	6.65 × 10^1^	5.79 × 10^1^	3.56 × 10^−1^	1.90 × 10^−1^	2.57 × 10^−1^	2.49 × 10^−1^
F21	min	−5.10 × 10^0^	−1.03 × 10^1^	−1.03 × 10^1^	−1.03 × 10^1^	−1.03 × 10^1^	−5.10 × 10^0^	−1.03 × 10^1^
	mean	1.00 × 10^0^	4.00 × 10^0^	4.00 × 10^0^	4.00 × 10^0^	4.00 × 10^0^	9.96 × 10^−1^	4.00 × 10^0^
	std	2.31 × 10^−4^	1.05 × 10^−9^	1.08 × 10^−5^	1.58 × 10^−3^	4.12 × 10^−3^	7.19 × 10^−3^	2.21 × 10^−2^
F22	min	−1.06 × 10^1^	−1.06 × 10^1^	−1.06 × 10^1^	−5.34 × 10^0^	−5.34 × 10^0^	−5.31 × 10^0^	−1.05 × 10^1^
	mean	4.00 × 10^0^	4.00 × 10^0^	4.00 × 10^0^	1.00 × 10^0^	1.00 × 10^0^	1.00 × 10^0^	3.99 × 10^0^
	std	7.03 × 10^−4^	6.73 × 10^−10^	7.83 × 10^−6^	8.32 × 10^−4^	1.53 × 10^−3^	1.94 × 10^−2^	1.85 × 10^−2^
F23	min	−1.07 × 10^1^	−5.19 × 10^0^	−1.07 × 10^1^	−5.19 × 10^0^	−5.19 × 10^0^	−5.32 × 10^0^	−1.02 × 10^1^
	mean	4.00 × 10^0^	6.00 × 10^0^	4.00 × 10^0^	6.00 × 10^0^	6.00 × 10^0^	9.98 × 10^−1^	4.01 × 10^0^
	std	2.01 × 10^−3^	1.22 × 10^−9^	9.56 × 10^−6^	1.10 × 10^−3^	9.11 × 10^−3^	2.23 × 10^−2^	2.86 × 10^−2^

**Table 7 biomimetics-09-00270-t007:** Comparison of different algorithms for solving the design problem of tension compression springs (100 iterations).

Algorithm	d	D	P	Cost
WOA	0.057197	0.50426	5.9959	0.01319
ACO	0.058969	0.55289	5.2814	0.013999
ALO	0.062969	0.68714	3.4902	0.014959
GWO	0.05	0.317381	14.0454	0.012731
GJO	0.055832	0.45237	7.5607	0.013482
SCGJO	0.05	0.316472	14.3817	0.012961
SCMGJO	0.052355	0.37238	10.4605	0.012719

**Table 8 biomimetics-09-00270-t008:** Comparison of different algorithms for solving the design problem of tension compression springs (300 iterations).

Algorithm	d	D	P	Cost
WOA	0.0500839	0.313818	14.6824	0.013132
ACO	0.053635	0.40164	9.283	0.013036
ALO	0.063608	0.717	3.1884	0.015051
GWO	0.05	0.315994	14.2685	0.012852
GJO	0.055007	0.44144	7.6696	0.012916
SCGJO	0.0510244	0.340295	12.4931	0.01284
SCMGJO	0.0500092	0.317531	14.0302	0.01273

**Table 9 biomimetics-09-00270-t009:** Comparison of different algorithms for solving the design problem of the three-bar truss (100 iterations).

Algorithm	*x* _1_	*x* _2_	Cost
WOA	0.78141	0.4292	263.9359
ACO	0.78153	0.42885	263.9344
ALO	0.88575	0.1855	269.0788
GWO	0.79052	0.40315	263.9088
GJO	0.78294	0.4257	264.0194
SCGJO	0.83404	0.29328	265.2301
SCMGJO	0.78981	0.40517	263.9087

**Table 10 biomimetics-09-00270-t010:** Comparison of different algorithms for solving the design problem of the three-bar truss (500 iterations).

Algorithm	*x* _1_	*x* _2_	Cost
WOA	0.79165	0.39989	263.903
ACO	0.78623	0.41531	263.9108
ALO	0.75445	0.51481	264.8722
GWO	0.78221	0.42687	263.9285
GJO	0.79267	0.39706	263.9086
SCGJO	0.77453	0.45173	264.2439
SCMGJO	0.78634	0.41488	263.8999

**Table 11 biomimetics-09-00270-t011:** Parameters of the threat source (Scene 1).

Number	Coordinates	Height	Radius
1	(400, 600, 0)	100	80
2	(600, 250, 0)	150	70
3	(500, 450, 0)	100	80
4	(350, 300, 0)	100	70
5	(700, 450, 0)	100	70
6	(650, 660, 0)	100	80

**Table 12 biomimetics-09-00270-t012:** Parameters of the threat source (Scene 2).

Number	Coordinates	Height	Radius
1	(400, 600, 0)	150	80
2	(600, 200, 0)	150	90
3	(500, 420, 0)	150	80
4	(300, 350, 0)	150	100
5	(700, 450, 0)	150	70
6	(150, 500, 0)	150	80
7	(350, 750, 0)	150	60
8	(800, 300, 0)	150	90
9	(600, 600, 0)	150	90

## Data Availability

Data are contained within the article.

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
