# Peer review of "A Multi-Objective Optimization Problem Solving Method Based on Improved Golden Jackal Optimization Algorithm and Its Application"

_biomimetics, 2024, doi:10.3390/biomimetics9050270_

Round 1

Reviewer 1 Report

Comments and Suggestions for Authors

In 12 this paper, a novel golden jackal optimization algorithm (SCGJO) combining sine and cosine and 13 Cauchy mutation is proposed.

The paper is very well strucutured and presented.

Through the optimization experiment of 23 benchmark test functions, the results show that the SCGJO algorithm performs well in convergence speed and accuracy. In addition, the stretching/compression spring design problem, three-bar truss design problem, and unmanned aerial vehicle path planning problem are introduced for verification. The experimental results prove that the SCGJO algorithm has superior performance compared with other intelligent optimization algorithms, and verifies its application ability in engineering applications.

Comments on the Quality of English Language

English is fine.

Author Response

Dear Editor,

Thank you for allowing a resubmission of our manuscript, with an opportunity to address the reviewers’ comments.

We are uploading our point-by-point response to the comments (below) (response to reviewers).

Reviewer 2 Report

Comments and Suggestions for Authors

The paper presents a sine-cosine and Cauchy mutation golden jackal optimization algorithm (SCGJO). Unfortunately, the paper in its current form is a little bit too long for an article, and it is not totally clear if it is meant to be a general presentation of the SCGJO algorithm versus other optimization algorithms or if it is meant to be a path planning optimization paper of UAVs as the title would suggest. 

The papers present the testing of the SCGJO algorithm against 23 benchmark functions and 3 test problems of which one is the path planning of UAVs. However, the proposed SCGJO algorithm is not a novel one. The sine-cosine and Cauchy mutation golden jackal optimization algorithm (SCGJO) has been already presented by Jinzhong Zhang, Gang Zhang, Min Kong and Tan Zhang in "SCGJO: A hybrid golden jackal optimization with a sine cosine algorithm for tackling multilevel thresholding image segmentation" published in June 2023 in Multimed Tools Applications, vol. 83, 7681–7719. https://doi.org/10.1007/s11042-023-15812-0

The only novelty introduced by the authors in the SCGJO algorithm in comparison to what is presented in the above-mentioned paper is the Cauchy mutation function. Therefore, to highlight the contribution of the authors to the field of knowledge they should have made a comparison of the proposed SCGJO algorithm with the Cauchy mutation against the SCGJO algorithm without the Cauchy mutation and not against the classical GJO algorithm in the testing part of the paper (The testing with the 23  benchmark functions).

As I said the paper is a little bit too long and the authors could reduce its length by revising the paper:

In the introduction and the body of the paper, several paragraphs say the same thing about the algorithm name and how it works, it is a little bit redundant. 

The first sentence in the abstract lacks an action verb from the sentence: "In view of the limitations ...." but nothing happens in this view, it only presents the limitations. The sentence is too long, by the end of it you forgot what you wanted to say.

In Figure 1 flowchart and in a few sentences in the text instead of golden jackal it appears leopard or golden leopard. These small typing mistakes could lead the reader to the false conclusion that there is no new strategy presented only the change of the animal name from an already existing optimization strategy.

Comments on the Quality of English Language

The paper presents the testing of a sine-cosine and Cauchy mutation golden jackal optimization algorithm (SCGJO), however this is not a novel algorithm, the SCGJO algorithm was already presented and used by Jinzhong Zhang, Gang Zhang, Min Kong and Tan Zhang in "SCGJO: A hybrid golden jackal optimization with a sine cosine algorithm for tackling multilevel thresholding image segmentation" published in June 2023 in Multimed Tools Applications, vol. 83, 7681–7719. https://doi.org/10.1007/s11042-023-15812-0

In Figure 1 flowchart and in a few sentences in the text instead of golden jackal it appears leopard or golden leopard. These could lead the reader to the conclusion that there is no new strategy presented only the change of the animal name from an already existing optimization strategy.

Author Response

(The authors gave the same response as above.)

Reviewer 3 Report

Comments and Suggestions for Authors

The authors improve the performance of the golden jackal algorithm combining with Sine-Cosine and Cauchy mutation. The proposed algorithm is teseted on 23 benchmark test functions.

page 6, formula (13) is not written in a proper way, put comma or space between the columns

page 8, figure 1 is written leopard instead jackal?

the dimension of the test functions is too small up to 50. To cn show real algorithm performance the dimension of the functions need to be at list 100.

Author Response

(The authors gave the same response as above.)

Round 2

Reviewer 2 Report

Comments and Suggestions for Authors

As it seems that the paper is meant to be a general presentation of the SCGJO algorithm with Cauchy mutation versus other optimization algorithms, the title of the paper is not in line with its content. 

The authors do not reference paper [39] as previously presenting the sine-cosine golden jackal optimization algorithm (SCGJO), the authors only mention paper [39] with regard to the Tent Mapping Reverse Learning. This would create a false impression to the reader that the Sine-Cosine Algorithm is a novelty to the traditional GJO algorithm proposed by the authors of the paper under review, which is not true.

The only novelty introduced by the authors in the SCGJO algorithm in comparison to what has been already presented in the literature by paper [39] is the Cauchy mutation function. Therefore, to showcase the improvement proposed by the authors to the SCGJO algorithm they should have made a comparison of the SCGJO algorithm with the Cauchy mutation against the SCGJO algorithm without the Cauchy mutation and not against the traditional GJO algorithm (with regard to the testing with the 23 benchmark functions and the 3 test problems). In its current form of the paper, it is not clear that the improvements as obtained results are due to the Sine-Cosine Algorithm (which is not a novelty proposed by the authors) or the Cauchy mutations (which is the only novelty proposed by the authors).  

Comments on the Quality of English Language

Some wording suggenstions: 

1. The first sentence of the Abstract is missing something: "In view of " what?.  I would suggest to give up the "In view of" beginning and start the sentence directly with "The traditional golden jackal ...."

2. At line 14 in the Abstract you should use "On one hand" instead of "On the one hand".

3. Table 1 on page 13 should be Table 5.

Author Response

Thank you for allowing a resubmission of our manuscript, with an opportunity to address the reviewers’ comments.

We are uploading our point-by-point response to the comments (below) (response to reviewers).

Reviewer 3 Report

Comments and Suggestions for Authors

All reviewers comments are taken in to account. The paper can be accepted as it is.

Author Response

Thank you for allowing a resubmission of our manuscript, with an opportunity to address the reviewers’ comments.

Thank you very much for the reviewer's review, which has greatly improved the paper.